# DEFT SCHEDULING OF DYNAMIC CLOUD WORKFLOWS WITH VARYING DEADLINES VIA MIXTURE-OF-EXPERTS

**Ya Shen**[∗]**, Gang Chen, Hui Ma & Mengjie Zhang**
School of Engineering and Computer Science, Victoria University of Wellington
Wellington, New Zealand
`{ya.shen, aaron.chen, hui.ma, mengjie.zhang}@ecs.vuw.ac.nz`

## ABSTRACT

Workflow scheduling in cloud computing demands the intelligent allocation of dynamically arriving, graph-structured workflows with varying deadlines onto ever-changing virtual machine resources. However, existing deep reinforcement learning (DRL) schedulers remain limited by rigid, single-path inference architectures that struggle to handle diverse scheduling scenarios. We introduce **DEFT** (**D**eadline-p**E**rceptive Mixture-o**F**-Exper**t**s), an innovative DRL policy architecture that leverages a specialized mixture of experts, each trained to manage different levels of deadline tightness. To our knowledge, DEFT is the first to introduce and validate a Mixture-of-Experts architecture for dynamic cloud workflow scheduling. By adaptively routing decisions through the most appropriate experts, DEFT is capable of meeting a broad spectrum of deadline requirements that no single expert can achieve. Central to DEFT is a **graph-adaptive** gating mechanism that encodes workflow deadlines and DAGs, task states, and VM conditions, using cross-attention to guide expert activation in a fine-grained, deadline-sensitive manner. Experiments on dynamic cloud workflow benchmarks demonstrate that DEFT significantly reduces execution cost and deadline violations, outperforming multiple state-of-the-art DRL baselines.

## 1 INTRODUCTION

As cloud computing provides elastic and on-demand computation resources, it has become a fundamental platform for running large-scale applications efficiently (Buyya & Broberg, 2011). Many of these applications take the form of workflows consisting of interdependent tasks, naturally modeled as *directed acyclic graphs* (DAGs) where nodes represent tasks and edges define precedence constraints. Each workflow is associated with a *service-level agreement* (SLA) deadline, missing which incurs financial penalties (Buyya et al., 2011; Shen et al., 2024). In practice, workflows arrive unpredictably, exhibit diverse structures, and have deadlines with varying levels of tightness, reflecting a broad range of user expectations. This work tackles the *Cost-Aware Dynamic Workflow Scheduling* (CADWS) problem, aiming to minimize total execution cost by jointly minimizing *virtual machine* (VM) rental fees and deadline violation penalties under dynamic and uncertain conditions. The diagram of CADWS is shown in Figures 1 (a) and (b).

Tackling the CADWS problem requires more than reactive scheduling. It demands intelligent policies that can reason under uncertainty and make fine-grained, deadline-aware decisions in real time. The core challenge lies in navigating a highly dynamic environment marked by fluctuating VM availability, diverse workflow structures, and, most critically, the *wide spectrum of deadline tightness* that governs *workflow urgency*. Schedulers must reason over complex task graphs, interpret shifting system states, and allocate resources in real time. These requirements quickly exceed the capabilities of traditional heuristics, which struggle to adapt to such temporal and structural changes.

Deep reinforcement learning (DRL) has shown growing promise for dynamic workflow scheduling in cloud environments (Zhou et al., 2024a; Ngwu et al., 2025). By modeling the scheduler as an

---

∗Corresponding author (ya.shen@ecs.vuw.ac.nz)

agent that interacts with a changing environment, DRL learns policies that optimize long-term cost and performance. In the CADWS setting, the agent observes system states and selects execution targets from a dynamically evolving VM pool. DRL schedulers have demonstrated clear advantages over heuristic methods in cloud workflow orchestration tasks (Shen et al., 2024; Yang et al., 2025), offering improved adaptability to structural variation and timing uncertainty.

Despite recent progress, a *key issue* remains in DRL-based CADWS approaches: the *inflexibility* of their *policy network architectures*. Most existing methods adopt a monolithic design, typically based on a single feedforward pathway (Huang et al., 2022; Shen et al., 2024). Once trained, these policies encode a static set of decision rules that are applied uniformly across different scheduling scenarios. While this can be effective in stable settings, such rigid architectures struggle to accommodate the wide variability in workflow deadline requirements, ranging from lenient to extremely tight. This lack of deadline awareness severely limits the scheduler's ability to allocate tasks appropriately under time pressure.

To overcome the limitations of rigid policy architectures, we propose **DEFT** (**D**eadline-p**E**rceptive Mixture-o**F**-Exper**t**s), a novel policy network tailored for dynamic workflow scheduling. Instead of relying on a single fixed inference pathway, DEFT dynamically selects from a pool of specialized subnetworks (experts), each trained to handle different levels of deadline tightness. Inspired by the Mixture-of-Experts (MoE) architecture in large language models (Shazeer et al., 2017), DEFT reinterprets expert activation as adaptive policy modulation, enabling fine-grained scheduling strategies as workflow urgency varies. It enhances the flexibility and responsiveness of DRL schedulers by diversifying action-priority mapping through three key innovations:

- **First MoE method for dynamic workflow scheduling.** DEFT is the first approach to bring MoE architectures into dynamic workflow scheduling, introducing a new level of adaptivity in deadline-aware policy design. Instead of relying on a single, rigid inference pathway, DEFT dynamically activates specialized experts based on the tightness of workflow deadlines. This design enables scalable, deadline-sensitive scheduling that conventional DRL schedulers fail to achieve.

- **Enhancing Policy Diversity and Generalization.** DEFT employs a novel two-phase training strategy: in the first phase, each expert is trained independently to specialize in a specific level of deadline tightness, learning tailored scheduling behaviors across the deadline spectrum. In the second phase, a graph-adaptive gating network is trained to dynamically route decisions through the most appropriate experts based on real-time workflow states, DAG structure, and VM conditions. During this phase, all experts are further fine-tuned alongside the gating network to ensure consistent and generalizable performance across a wide range of scheduling contexts.

- **Graph-adaptive Gating.** DEFT is powered by a novel graph-adaptive gating network that uses cross attention to integrate structured workflow representations with real-time scheduling context. This mechanism enables fine-grained, deadline-aware expert activation at each decision step, allowing the policy to fluidly adapt to changing deadlines and resource conditions. To the best of our knowledge, our gating design is the first to combine graph neural representations with MoE routing in a DRL scheduler, offering a principled and scalable approach to structured, deadline-driven scheduling in dynamic cloud environments.

## 2 RELATED WORK

We begin by reviewing recent advances in DRL for CADWS, followed by an overview of MoE architectures and their integration into DRL policy networks.

### 2.1 DEEP REINFORCEMENT LEARNING FOR CADWS

DRL has shown strong potential in learning effective scheduling policies by leveraging the representation power of neural networks. Early DRL-based schedulers were developed for Job-Shop Scheduling (JSS) (Zhang et al., 2020; Song et al., 2022) and Vehicle Routing Problems (VRP) (Wu et al., 2021; Xin et al., 2021), and have since been extended to the more complex CADWS setting (Huang et al., 2022; Jayanetti et al., 2024). These early CADWS studies demonstrated DRL's

superiority over heuristic methods, though they relied on a simple FFN as the policy network, limiting policy expressiveness. Subsequent CADWS works improved policy network design by introducing self-attention (Shen et al., 2024) and GNNs (Shen et al., 2025; Yang et al., 2025) to capture task dependencies in workflows.

As shown in Figure 1 (c), the policy networks in these studies often comprise two key modules: a **State Embedding Module (SEM)** that encodes raw environment states into informative state embeddings, and a **Priority Mapping Module (PMM)** that further maps these embeddings to the action priorities for VM selection. Current CADWS studies mainly focus on SEM design, while PMMs are typically implemented as a single Feed-Forward Network (FFN) with a fixed inference pathway, which limits their ability to adapt to varying deadline tightness. This raises two key questions: (1) Can a fixed-pathway PMM fully leverage rich state embeddings to capture the diverse scheduling needs imposed by varying deadline tightness? (2) Would a set of specialized inference pathways provide stronger adaptability to dynamic workflow deadlines and resource conditions? To answer these questions, we propose DEFT, a novel policy network that replaces the monolithic PMM with a MoE architecture. DEFT dynamically selects expert pathways based on workflow urgency, enhancing flexibility and generalization for deadline-sensitive cloud scheduling.

## 2.2 EVOLUTION OF MIXTURE-OF-EXPERTS (MoE)

The MoE paradigm was first introduced by Jacobs et al. (1991), where a gating network assigns inputs to specialized expert networks. More recently, MoE architectures have been widely adopted in large-scale learning tasks, demonstrating their ability to improve model expressiveness and context sensitivity through dynamic expert selection (Shazeer et al., 2017; Fedus et al., 2022; Du et al., 2022).

Applying MoE to DRL for combinatorial optimization remains relatively underexplored. Prior work by Kidambi et al. (2020) showed that MoE can enhance sample efficiency and generalization capability of DRL agents. For example, recent studies have adopted MoE, such as MVMoE (Zhou et al., 2024b) and SHIELD (Goh et al., 2025), to solve multiple vehicle routing variants. Nevertheless, these early efforts are not well suited for CADWS. First, they focus on static settings, whereas CADWS involves continuously arriving workflows with varying deadline urgency. Second, their gating mechanisms are implemented by simple linear layers, unable to leverage the rich DAG structures and dynamic contexts needed for effective expert selection. We address these gaps with a novel MoE-based policy network tailored for CADWS, featuring a graph-adaptive gating module that adaptively routes decisions based on workflow topology, task states, and deadline tightness.

## 3 PRELIMINARIES

This section defines the CADWS problem, presents its Markov Decision Process (MDP) formulation, and specifies the optimization objectives.

### 3.1 COST-AWARE DYNAMIC WORKFLOW SCHEDULING

The CADWS problem aims to schedule a set of dynamically arriving workflows $\mathcal{W}$ for execution on VMs in a cloud environment. Each workflow $W_i \in \mathcal{W}$ is represented by a DAG $W_i = (\mathcal{O}_{W_i}, \mathcal{C}_{W_i})$, where $\mathcal{O}_{W_i}$ is the set of computational tasks and $\mathcal{C}_{W_i}$ encodes precedence constraints as directed edges. Any directed edge $(O_{ni}, O_{nj}) \in \mathcal{C}_{W_i}$ indicates that task $O_{ni}$ must be completed before task $O_{nj}$ can begin. A task with all its predecessor tasks completed is considered the *ready task*, denoted as $O_{n*}$, and is eligible for immediate execution.

Tasks exhibit heterogeneous workloads. Each task $O_{ni} \in \mathcal{O}_{W_i}$ has a computational demand $cd_{O_{ni}} \in \mathbb{R}^+$. Workflows arrive dynamically across time, each with an arrival time $a_i$ and a deadline $d_i$ derived from a user-specified SLA. To model the varied urgency levels across workflows, we define each workflow's deadline as:

$$d_i = a_i + \gamma \cdot \text{minMakespan}(W_i), \tag{1}$$

where $a_i$ is the arrival time of workflow $W_i$, $\gamma \geq 1$ is a *deadline relaxation coefficient* that controls the *tightness* of the deadline, and minMakespan($W_i$) denotes the minimum execution time achievable by allocating the fastest available VM to each task of $W_i$ without any delay. Smaller $\gamma$ values

lead to tighter deadlines, posing greater challenges for the scheduler to meet timing constraints under dynamic resource availability.

The cloud infrastructure offers a pool of VMs $\mathcal{M} = \{m_1, m_2, \ldots, m_{|\mathcal{M}|}\}$, whose availability may change over time. Each VM $m_j \in \mathcal{M}$ has a processing speed $v_j$ and an hourly rental cost $c_j$. Under a pay-as-you-go model (Ibrahim et al., 2011), VMs can be provisioned on demand without predefined capacity constraints, enabling flexible but cost-sensitive resource allocation. A more detailed description of the problem definition can be found in Appendix A.

### 3.2 Markov Decision Process Formulation

We model the CADWS problem as an undiscounted Markov Decision Process (MDP) defined by the tuple $(\mathcal{S}, \mathcal{A}, \Pr, \mathcal{R})$. Each of its elements is introduced below.

**State Space $\mathcal{S}$:** At any time step $t$, the state $s_t \in \mathcal{S}$ captures the full system status, including: (1) *Workflow information*: workflow DAGs, arrival times, deadlines, task completion status, and workloads; and (2) *VM information*: current VM instances, their types, processing speeds, queue lengths, and existing lease time.

**Action Space $\mathcal{A}$:** At each time step $t$, the action $a_t \in \mathcal{A}$ specifies the assignment of the current ready task $n^*$ to a VM instance for execution. The action space is dynamic and consists of two types of options: (1) assignment to an active (already leased) VM, and (2) provisioning and assignment to a new VM of any available type. This flexible formulation allows the DRL scheduler to dynamically lease new VMs on demand, enabling adaptive resource scaling throughout the scheduling process.

**Transition Probability $\Pr$:** The environment transition $\Pr(s_{t+1}|s_t, a_t)$ captures the stochastic evolution of workflow arrivals, VM availability, and task execution. The transition model is unknown to the DRL scheduler.

**Reward Function $\mathcal{R}$:** The reward in CADWS is derived from two sources of costs over the scheduling horizon $T$. At each time step $t$, the scheduler incurs an *immediate VM rental cost* $C_t^{\mathrm{vm}} \geq 0$ for leasing all active VM instances. In addition, an *episodic SLA penalty* $C_T^{\mathrm{sla}}(\mathcal{W}) \geq 0$ is computed at the final time step to quantify the total penalty incurred by workflows that fail to meet their deadlines. In line with these cost components, we define the total return (i.e., negative total cost) for a trajectory $\tau$ as:

$$R(\tau) = -\sum_{t=0}^{T-1} C_t^{\mathrm{vm}} - C_T^{\mathrm{sla}}(\mathcal{W}). \tag{2}$$

with the VM rental cost calculated as:

$$C_t^{\mathrm{vm}} = c_j \cdot \left\lceil \frac{cd_{O_{n^*}}}{v_j \cdot 3600} \right\rceil, \tag{3}$$

where $c_j$ is the hourly cost of VM $m_j$ assigned to task $O_{n^*}$, and $v_j$ is its processing speed. Meanwhile, the total SLA penalty is computed as the sum over all workflows:

$$C_T^{\mathrm{sla}}(\mathcal{W}) = \sum_{W_i \in \mathcal{W}} C_T^{\mathrm{sla}}(W_i), \tag{4}$$

$$C_T^{\mathrm{sla}}(W_i) = \beta \cdot \max\{0, \mathrm{CT}(W_i) - d_i\}, \tag{5}$$

where $\beta$ is the penalty coefficient. $\mathrm{CT}(W_i)$ is the actual completion time of workflow $W_i \in \mathcal{W}$, and $d_i$ is the workflow's SLA deadline. Let $J(\pi) = \mathbb{E}_{\tau \sim \pi}[R(\tau)]$ be the expected total return under policy $\pi$. The goal of the DRL scheduler is to learn an optimal policy $\pi^*$ that minimizes the total scheduling cost, or equivalently, maximizes the overall return:

$$\pi^* = \arg\max_{\pi} J(\pi). \tag{6}$$

## 4 Methodology

This section introduces our DEFT approach in detail. Figure 1 sketches the overall framework of DEFT. Figure 2 elaborates on the internal modules of DEFT. Algorithms in Appendices C and D present a two-phase strategy for training the DEFT policy.

## 4.1 THE PROPOSED DEFT POLICY

As illustrated in Figure 1 (c), DEFT enhances policy expressiveness by replacing the fixed policy mapping module (PMM) with an MoE architecture (Jacobs et al., 1991). Instead of relying on a single inference pathway, DEFT trains a set of sparse sub-networks or experts, each specialized for a different level of deadline tightness. At inference time, a graph-adaptive gating network dynamically selects the most suitable expert based on the current scheduling context.

Figure 2 provides a detailed view: panel (A) shows how MoE diversifies the inference pathway to support varied policy behaviors, while panel (B) illustrates our novel gating design that leverages workflow DAG structure, task states, and deadline urgency to guide expert activation. This integration empowers DEFT to flexibly align its scheduling strategy with a broad spectrum of dynamic, deadline-driven scenarios.

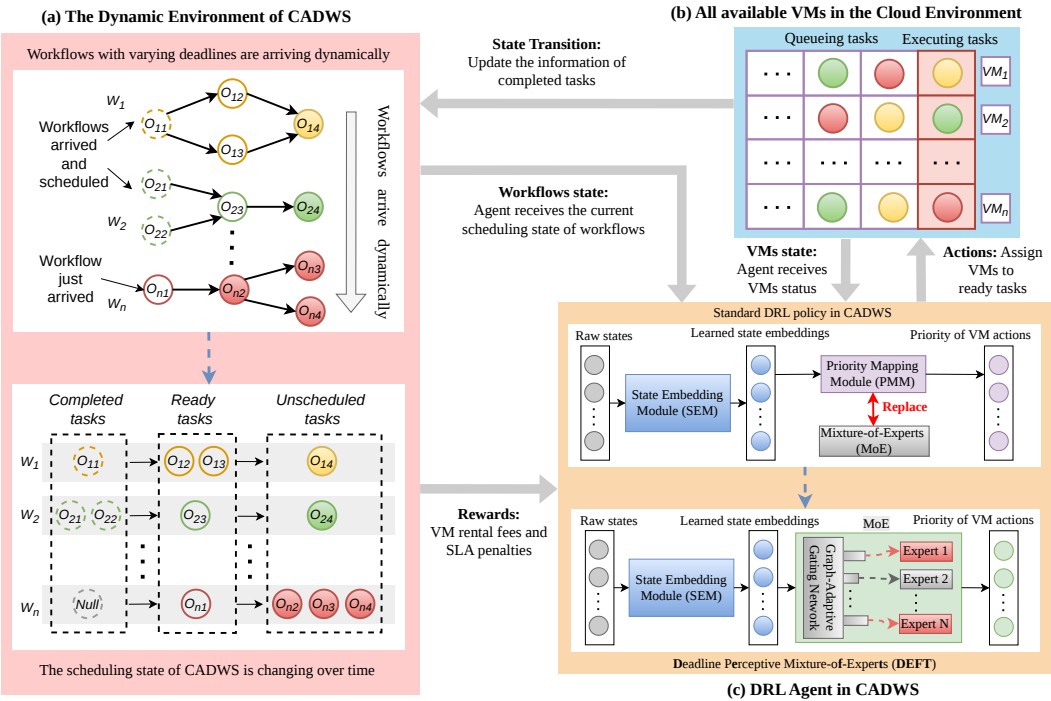

Figure 1: The scheduling of dynamic workflows via DRL. (a) Workflows arrive over time, each associated with a distinct deadline; (b) The set of available VMs in the cloud fluctuates over time; and (c) The DRL agent selects appropriate VM resources for task execution by observing workflow and VM states. In this work, our main contribution is to enhance the DRL policy network by incorporating a Mixture-of-Experts (MoE) architecture, enabling more intelligent decision-making.

## 4.2 DEADLINE-AWARE EXPERT DESIGN

MoE architectures are often used to partition the dense layers in neural networks into multiple lightweight MLP-based experts, each trained to handle different sub-tasks. In **DEFT**, we tailor this paradigm for CADWS by assigning each expert to a particular *deadline tightness regime*.

Let $\gamma \in \Gamma$ denote a discrete set of deadline relaxation coefficients (e.g., $\Gamma = \{1.25, 1.75, 2.25, 5.0\}$), which control the slackness of workflow deadlines. For each $\gamma_i \in \Gamma$, we instantiate a corresponding expert $\mathrm{EXP}_i$ and independently pre-train it using only workflows whose deadlines are generated with that specific $\gamma_i$. This enables each expert to learn scheduling behaviors optimal for its respective urgency level, ranging from aggressive early scheduling under tight deadlines to cost-efficient delay-tolerant strategies under relaxed ones.

After expert pretraining, we freeze the expert structure and jointly fine-tune all experts along with a dedicated *graph-adaptive gating network*. This gating network takes as input: (1) the global workflow embedding produced by the SEM module, (2) the current task node embedding and its

topological context (e.g., predecessors, successors), (3) system-level features such as VM availability, and (4) a normalized deadline tightness score. These inputs are fused via cross-attention layers to compute a sparse routing vector over the expert set, enabling the selection of the most appropriate expert at each decision step.

As illustrated in Figure 2 (A), this design allows DEFT to dynamically adapt its inference pathway based on the urgency of each incoming workflow and the evolving system context. Crucially, expert activation is *not static per workflow* but evolves over time, facilitating fine-grained, deadline-aware scheduling across numerous workflows.

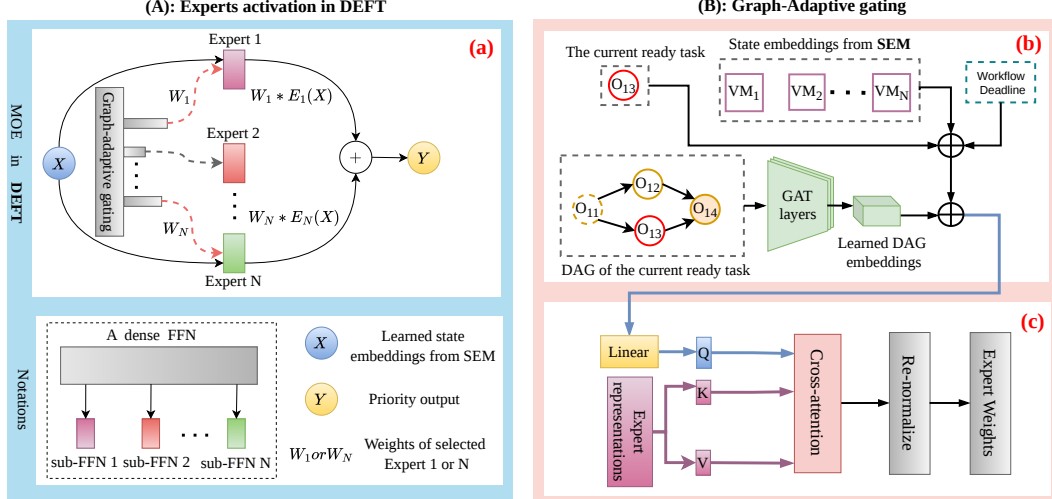

Figure 2: The MoE and graph-adaptive gating network in DEFT. (a) The SEM-generated VM embedding is routed to the top-$K$ experts selected by the gating network, and their weighted outputs are aggregated to produce the scheduling priority. (b) The gating network encodes workflow DAG structure along with VM states, ready task features, and deadline tightness to form the query vector $Q$, while expert representations act as keys $K$ and values $V$. (c) Cross-attention computes expert weights that guide expert selections.

## 4.3 GRAPH-ADAPTIVE GATING NETWORK

The effectiveness of DEFT in CADWS comes from its ability to activate the most suitable experts at each decision step, especially under dynamic workflows with heterogeneous deadlines. Since workflows are naturally represented as DAGs with complex task dependencies, effective expert routing must account for this structure. Conventional MoE gating networks, based on simple linear projections or shallow MLPs, cannot fully capture such topological and contextual information (Cai et al., 2025), making them inadequate for deadline-sensitive scheduling.

To address this challenge, we design a graph-adaptive gating network for fine-grained, context-sensitive expert selection. At each scheduling step, the workflow DAG is encoded using a graph attention network (GAT) (Veličković et al., 2017) to capture structural patterns and global dependencies. The resulting DAG embedding is then fused with workflow deadline features, and a cross-attention mechanism processes this context to weight and activate the most relevant experts. This design allows DEFT to dynamically route inference through the expert pathway best suited to the current scheduling scenario, as illustrated in Figure 2 (B).

**DAG Embedding Learning.** As shown in Figure 2 (B), we employ a GAT to capture global correlations and task dependencies within the workflow, producing informative DAG embeddings. GAT's attention mechanism dynamically weights neighboring task nodes, enabling the model to focus on the most critical inter-node dependencies. The DAG embeddings produced by the GAT modules are then used in a cross-attention module to select the specific experts to activate. The detailed DAG embedding learning process is described in Appendix B.1.

**Cross-Attention for Expert Selection.** As shown in Figure 2 (B), the gating network leverages a cross-attention mechanism to perform fine-grained, deadline-aware expert selection. Specifically, the Query (Q) is formed by concatenating four components: the learned DAG embedding, the feature representation of the current ready task, the VM state embeddings from the SEM, and the workflow's dynamic deadline information. The feature embeddings of all experts are set as both the Key (K) and Value (V) in the attention operation. The resulting attention scores are normalized to form a probability distribution over experts to guide expert selection.

This cross-attention design allows the gating network to make deeply contextualized expert selections. At each decision step, it integrates the workflow DAG, the current ready task, and most importantly deadline urgency to choose the suitable expert for VM action prioritization. In doing so, the gating network learns an expert-routing policy that leverages structural and temporal signals to deliver accurate, deadline-aware scheduling decisions. The cross-attention procedure is detailed in Appendix B.2.

### 4.4 TRAINING METHOD

Our proposed DEFT method adopts a two-phase pipeline to effectively train the MoE policy network. In the *Expert Pre-training* phase, multiple experts are individually pre-trained to tackle specialized scheduling scenarios with varied deadline tightness. In the subsequent *Gating Network Training* phase, these experts are integrated into DEFT and further improved together with the gating module and the SEM module to make adaptive end-to-end scheduling decisions in dynamic cloud environments.

#### 4.4.1 EXPERT PRE-TRAINING

To ensure that each expert in DEFT acquires diversified expertise for varied deadline settings, we pre-train the policy network with multiple different $\gamma$, e.g., $\gamma \in \{1.25, 1.75, 2.25, 5.0\}$. For each setting of $\gamma$, the policy network is trained via OpenAI-ES (Salimans et al., 2017) until convergence. Afterwards, the trained policy parameters are extracted and stored. These pre-trained parameters enable us to establish multiple experts. Each expert is initialized with knowledge specific to a class of deadline tightness before being integrated into the full DEFT architecture. Appendix C gives the detailed training steps of each expert.

#### 4.4.2 GATING NETWORK TRAINING

After the above phase, all pre-trained experts are loaded into the DEFT policy to form the expert pool. Subsequently, both the gating network and the SEM module are trained simultaneously together with all the experts. The gating network routes tasks to the most suitable expert in line with the workflow DAG and deadline, while the SEM refines the representation of context-aware scheduling states. In this phase, we continue to fine-tune experts for enhanced adaptability. Meanwhile, the gating network is trained to accurately identify and select the most suitable experts to handle the respective workflow deadlines. This hybrid training strategy enables DEFT to combine specialized expertise with adaptive gating, delivering robust scheduling across diverse and dynamic deadline scenarios. The detailed algorithm for training the gating network can be found in Appendix D.

## 5 EXPERIMENTS

This section evaluates DEFT's performance, starting with the experimental setup and baseline methods, followed by comprehensive comparisons under dynamic workflow scenarios to examine DEFT's key components. The code is available at https://github.com/yashenCS/DEFT.

### 5.1 PROBLEM SETTINGS AND EXPERIMENT CONFIGURATION

**Workflow in CADWS.** We evaluate the proposed **DEFT** method on the CADWS problem using a widely adopted simulator (Shen et al., 2025; 2024; Huang et al., 2022). The simulator models heterogeneous VM instances (detailed in Appendix E) and four representative workflow patterns: CyberShake, Montage, Inspiral, and SIPHT (Deelman et al., 2015). Workflows are categorized into small (S), medium (M), and large (L) scales according to the number of tasks per workflow, thereby

reflecting a wide range of scheduling complexities. Workflow arrivals are generated using a Poisson process with rate $\lambda = 0.01$, which captures the dynamic and stochastic nature of real-world cloud environments (Huang et al., 2022; Shen et al., 2024). The SLA deadline of workflows is governed by two coefficients: $\gamma$ for deadline tolerance and $\beta = 0.24$/hour (Shen et al., 2024) for penalty severity. Larger $\gamma$ values relax deadlines, while larger $\beta$ values amplify penalty costs. They together demand scheduling policies to strike a balance between renting cheaper VMs and avoiding deadline violations.

**Baselines.** We experimentally compare DEFT with five baselines, including **ProLis** (Wu et al., 2017) and **GRP-HEFT** (Faragardi et al., 2019) as popular priority-based heuristic approaches, as well as **ES-RL** (Huang et al., 2022), **SPN-CWS** (Shen et al., 2024), and **GATES** (Shen et al., 2025) as state-of-the-art DRL techniques for CADWS. Particularly, SPN-CWS adopts a Transformer-based policy network. GATES uses GNNs to model its trained policy networks. Since GATES has shown cutting-edge performance on CADWS, DEFT builds on its neural network architecture and directly inherits its GNN-based policy network as its **SEM** module, as illustrated in Figure 1 (c).

**Parameters of DEFT.** To demonstrate the reliability of DEFT, we directly adopt the hyperparameter settings recommended in GATES (Shen et al., 2025) without additional fine-tuning. We construct the graph-adaptive gating network in DEFT with two GAT layers. OpenAI-ES (Salimans et al., 2017) is utilized to train the DEFT policy. This algorithm uses a population size of 40, 3000 generations, an initial learning rate of 0.01, and Gaussian noise with a standard deviation of 0.05.

**Two-phase training.** (*Phase-1: Expert pre-training*) We train four experts on **S-scale** instances under fixed deadlines $\gamma \in \{1.25, 1.75, 2.25, 5.0\}$, each expert specializing in a single deadline regime. Every training instance contains 10 workflows with identical $\gamma$. OpenAI-ES evaluates *one instance per generation* with Poisson arrivals ($\lambda = 0.01$). (*Phase-2: Gating network training with expert fine-tuning*) Starting from the pre-trained experts, we jointly optimize the SEM, gating network, and experts on S-scale instances. For each training instance, the deadline is sampled from $\gamma \in \{1.0, 1.25, 1.5, 1.75, 2.0, 2.25, 3.0\}$, ensuring that the gating network learns to adapt expert selection to varying deadline tightness. Additionally, all baselines are trained on the same mixed-deadline dataset as DEFT's stage-2 training. This ensures that all methods learn under the same data distribution.

**Testing.** We test on **S/M/L** scales with **30 instances** per scale; each instance contains 10 workflows with their $\gamma$ sampled from the same set as above. This setting evaluates DEFT under instances with varying deadline regimes and its generalization from small-scale workflows to larger-scale workflows. All testing is based on 10 independent runs. Additional details are provided in Appendix F.

## 5.2 MAIN RESULTS UNDER DYNAMIC DEADLINES

**Total cost.** Table 1 reports results where each test instance is assigned a different deadline, aiming to evaluate the algorithm's scheduling performance under highly varying deadline conditions. DEFT has the lowest total cost at all sizes (S/M/L: 52.46 / 86.60 / 137.69). The margin over the baseline (GATES) grows with scale: S improves by 0.49 (52.95→52.46; 0.9%), M by 11.16 (97.76→86.60; 11.4%), and L by 57.96 (195.65→137.69; 29.6%). ES-RL degrades sharply with scale (65.39→225.46). SPN-CWS beats ES-RL at M (87.69 vs. 109.23). In short: DEFT achieves strong scalability since its total cost rises more slowly as workflows grow and deadlines tighten. In contrast, heuristic schedulers such as ProLis and GRP-HEFT perform much worse across all scales, showing costs three to ten times higher than DRL-based approaches, indicating their inability to cope with dynamic deadlines.

**VM/SLA balance.** By observing Table 1, ProLis suffers high penalties despite moderate VM fees, while GRP-HEFT avoids violations by overusing costly VMs, making both inferior to DRL methods. In S, GATES excels at minimizing VM fees (20.65) and DEFT focuses more on minimizing SLA (31.45); DEFT's slightly higher VM (21.01) is offset by its lower SLA, yielding superior total cost (52.46 vs. 52.95). In M and L, DEFT shifts to cutting VM most (41.06, 70.88), while GATES aims to reduce SLA penalty (42.42, 50.45). Even though DEFT's SLA is not minimal in M/L (45.54, 66.81), the VM savings dominate, so its total cost still leads the competition (86.60 vs. 97.76; 137.69 vs. 195.65). The takeaway is prominent: minimizing one component (VM or SLA) is insufficient; DEFT adapts the trade-off with scale and wins on the overall cost.

**Generalization:** Since all DRL methods are trained on the same S-scale data with identical mixed-deadline settings, their performance on the S-scale test set is naturally similar. DEFT still achieves a consistent improvement in this in-distribution setting, but its primary advantage appears when evaluated on M and L scales. The markedly larger gains on these unseen scales highlight DEFT's stronger generalization ability, which is the intended benefit of its expert specialization and graph-adaptive gating design.

Table 1: Total cost (mean) with VM fees / SLA penalties under dynamic deadlines.

| Scenario | ProLis | | GRP-HEFT | | ES-RL | | SPN-CWS | | GATES | | DEFT | |
|---|---|---|---|---|---|---|---|---|---|---|---|---|
| | Cost | VM/SLA | Cost | VM/SLA | Cost | VM/SLA | Cost | VM/SLA | Cost | VM/SLA | Cost | VM/SLA |
| $\langle S \rangle$ | 183.74 | 79.01 / 104.73 | 297.58 | 297.58 / **0.00** | 65.39 | 29.12 / 36.27 | 54.99 | 23.31 / 31.68 | 52.95 | **20.65** / 32.30 | **52.46** | 21.01 / 31.45 |
| $\langle M \rangle$ | 304.03 | 176.34 / 127.69 | 495.64 | 495.64 / **0.00** | 109.23 | 63.75 / 45.48 | 87.69 | 44.05 / 43.64 | 97.76 | 55.34 / 42.42 | **86.60** | **41.06** / 45.54 |
| $\langle L \rangle$ | 755.21 | 279.43 / 475.78 | 1064.34 | 1064.34 / **0.00** | 225.46 | 170.45 / 55.01 | 149.26 | 90.64 / 58.62 | 195.65 | 145.20 / 50.45 | **137.69** | **70.88** / 66.81 |

## 5.3 TEST PERFORMANCE ON SCENARIOS WITH DIFFERENT DEADLINES

**Total cost comparison.** Table 2 fixes the deadline across all test instances to evaluate scalability with workflow size. DEFT consistently achieves the lowest total cost, while heuristic schedulers (ProLis, GRP-HEFT) either overspend on VMs or incur large penalties. ES-RL and SPN-CWS perform less stably, confirming DEFT's superiority. In contrast, DEFT handles both tight and relaxed deadlines well, yielding robust cost reduction beyond single-expert methods. This advantage comes directly from DEFT's MoE design: the graph-adaptive gating network can select the most suitable expert in line with the current deadline levels, allowing DEFT to minimize total cost across diverse scenarios.

**VM/SLA balance.** As shown in Table 2, the VM and SLA results reveal distinct biases among existing algorithms: GRP-HEFT meets deadlines at very high VM cost; ProLis incurs large SLA penalties; and SPN-CWS cuts VM usage aggressively but suffers frequent deadline violations. GATES achieves a more balanced trade-off, but DEFT surpasses it by delivering consistently lower costs. DEFT's MoE with graph-adaptive gating dynamically routes tasks to experts based on deadline tightness, simultaneously avoiding excessive VM usage and large SLA penalties. Its adaptive expert selection capabilities keep both VM cost and SLA penalties at a low level, resulting in the lowest total cost. More results regarding convergence and stability can be found in Appendix G.

Table 2: Total cost (mean) and VM rental fees / SLA penalties across every single deadline scenario.

| Scenario | ProLis | | GRP-HEFT | | ES-RL | | SPN-CWS | | GATES | | DEFT | |
|---|---|---|---|---|---|---|---|---|---|---|---|---|
| | Cost | VM/SLA | Cost | VM/SLA | Cost | VM/SLA | Cost | VM/SLA | Cost | VM/SLA | Cost | VM/SLA |
| $\langle 1.0, S \rangle$ | 133.74 | 83.25/50.49 | 222.90 | 222.90/**0.00** | 54.31 | 30.91/23.40 | 45.37 | 18.45/26.92 | 44.58 | 18.80/25.78 | **41.58** | **17.74**/23.84 |
| $\langle 1.0, M \rangle$ | 203.79 | 90.05/113.74 | 339.65 | 339.65/**0.00** | 86.37 | 58.72/27.65 | 72.86 | 36.14/36.72 | 67.93 | 35.43/32.50 | **66.48** | **33.89**/32.59 |
| $\langle 1.0, L \rangle$ | 311.70 | 179.90/131.80 | 519.50 | 519.50/**0.00** | 131.12 | 103.65/27.47 | 112.78 | 64.70/48.08 | 103.90 | 65.76/38.14 | **100.47** | **57.21**/43.26 |
| $\langle 1.25, S \rangle$ | 118.17 | 50.59/67.58 | 196.95 | 196.95/**0.00** | 47.33 | 30.52/16.81 | 40.58 | 17.74/22.84 | 39.39 | 17.79/21.60 | **37.43** | **17.34**/20.09 |
| $\langle 1.25, M \rangle$ | 190.62 | 108.34/82.28 | 317.70 | 317.70/**0.00** | 77.07 | 55.76/21.31 | 68.40 | 34.97/33.43 | 63.54 | 33.83/29.71 | **61.53** | **32.47**/29.06 |
| $\langle 1.25, L \rangle$ | 421.32 | 225.31/196.01 | 526.65 | 526.65/**0.00** | 122.89 | 98.00/24.89 | 112.37 | 65.98/46.39 | 105.33 | 68.72/36.61 | **99.16** | **57.43**/41.73 |
| $\langle 1.5, S \rangle$ | 152.44 | 95.68/56.76 | 190.55 | 190.55/**0.00** | 48.38 | 27.72/20.66 | 39.53 | 17.86/21.67 | 38.11 | 17.99/20.12 | **36.87** | **17.74**/19.13 |
| $\langle 1.5, M \rangle$ | 187.65 | 94.21/93.44 | 375.30 | 375.30/**0.00** | 80.33 | 52.42/27.91 | 65.68 | 34.18/31.50 | 62.56 | 34.05/28.51 | **61.37** | **32.95**/28.42 |
| $\langle 1.5, L \rangle$ | 306.09 | 210.33/95.76 | 510.15 | 510.15/**0.00** | 138.61 | 104.76/33.85 | 114.46 | 68.57/45.89 | 102.03 | 65.88/36.15 | **98.31** | **58.02**/40.29 |
| $\langle 1.75, S \rangle$ | 149.24 | 63.85/85.39 | 186.55 | 186.55/**0.00** | 52.68 | 30.34/22.34 | 39.14 | **18.42**/20.72 | 37.31 | 18.61/18.70 | **36.75** | 18.46/18.29 |
| $\langle 1.75, M \rangle$ | 187.17 | 122.46/64.71 | 374.34 | 374.34/**0.00** | 96.93 | 65.84/31.09 | 64.52 | 34.55/29.97 | 62.39 | 34.71/27.68 | **60.57** | **33.04**/27.53 |
| $\langle 1.75, L \rangle$ | 419.36 | 259.55/159.81 | 524.20 | 524.20/**0.00** | 212.38 | 171.14/41.24 | 109.96 | 66.40/43.56 | 104.85 | 69.41/35.44 | **98.20** | **59.46**/38.74 |
| $\langle 2.0, S \rangle$ | 96.12 | 40.72/55.40 | 192.24 | 192.24/**0.00** | 47.91 | 26.80/21.11 | 37.04 | 18.19/18.85 | 35.48 | 18.11/17.37 | **34.63** | **18.06**/16.57 |
| $\langle 2.0, M \rangle$ | 222.36 | 127.46/94.90 | 333.54 | 333.54/**0.00** | 91.11 | 61.82/29.29 | 59.37 | 32.12/27.25 | 57.69 | 32.37/25.32 | **56.94** | **31.88**/25.06 |
| $\langle 2.0, L \rangle$ | 265.92 | 159.12/106.80 | 443.20 | 443.20/**0.00** | 201.62 | 161.33/40.29 | 102.00 | 61.59/40.41 | 97.09 | 63.63/33.46 | **93.80** | **56.71**/37.09 |
| $\langle 2.25, S \rangle$ | 125.92 | 82.68/43.24 | 157.40 | 157.40/**0.00** | 44.50 | 25.16/19.34 | 35.31 | 17.95/17.36 | 34.79 | 18.36/16.43 | **33.01** | **17.75**/15.26 |
| $\langle 2.25, M \rangle$ | 165.45 | 79.98/85.47 | 330.90 | 330.90/**0.00** | 82.73 | 56.04/26.69 | 58.59 | 32.51/26.08 | 56.80 | 32.13/24.67 | **55.45** | **31.56**/23.89 |
| $\langle 2.25, L \rangle$ | 352.08 | 179.93/172.15 | 440.10 | 440.10/**0.00** | 201.48 | 162.72/38.76 | 103.54 | 63.74/39.80 | 95.95 | 63.32/32.63 | **92.55** | **56.25**/36.30 |
| $\langle 3.0, S \rangle$ | 97.44 | 59.49/37.95 | 194.88 | 194.88/**0.00** | 41.86 | 23.21/18.65 | 32.76 | 17.60/15.16 | 32.48 | 18.17/14.31 | **30.74** | **17.53**/13.21 |
| $\langle 3.0, M \rangle$ | 166.65 | 93.36/73.29 | 277.75 | 277.75/**0.00** | 76.03 | 50.10/25.93 | 55.57 | 32.23/23.34 | 55.54 | 33.34/22.20 | **52.32** | **31.44**/20.88 |
| $\langle 3.0, L \rangle$ | 293.10 | 157.89/135.21 | 488.50 | 488.50/**0.00** | 163.45 | 128.03/35.42 | 100.52 | 65.42/35.10 | 97.70 | 67.29/30.41 | **93.69** | **58.06**/35.63 |

## 5.4 ABLATION STUDIES

We evaluate DEFT on the same testing scenarios by analyzing its gating design, the effect of replacing the PMM of GATES with a deeper MLP, and its average per-step inference overhead. The results show that DEFT delivers the best overall performance while keeping inference overhead modest.

**Comparing Gating Mechanisms.** We first isolate the effect of the gating network inside DEFT. All gating networks receive the same input embedding; they differ only in how they score experts. As summarized in Table 3, the linear gating performs the worst, the MLP gating improves but still falls behind, and the graph-adaptive gating used by DEFT consistently achieves the lowest total cost on all S/M/L scales. This confirms that CADWS benefits from the proposed graph-adaptive gating that is aware of workflow structure and deadline pressure, and that simple linear or MLP gating cannot fully exploit expert specialization.

Table 3: Performance and average per-step inference overhead on different testing scales.

| Method | Total Cost | | | Average Inference Overhead (seconds/step) | | | |
|---|---|---|---|---|---|---|---|
| | S | M | L | S | M | L | Overall (S+M+L) |
| GATES (original PMM) | 52.95 | 97.76 | 195.65 | 0.0616 | 0.1610 | 0.4250 | **0.2159** |
| GATES + deep MLP-PMM | 52.91 | 98.41 | 194.77 | 0.0674 | 0.1267 | 0.6979 | 0.2973 |
| DEFT + Linear gating | 52.85 | 88.41 | 142.27 | 0.0608 | 0.1453 | 0.4467 | 0.2176 |
| DEFT + Graph-adaptive gating (ours) | **52.46** | **86.60** | **137.69** | 0.0648 | 0.1482 | 0.4525 | 0.2218 |
| DEFT + MLP gating | 52.70 | 87.34 | 141.62 | 0.0777 | 0.1586 | 0.5206 | 0.2523 |

**MoE-PMM vs. MLP-PMM.** To check whether DEFT's gain over GATES comes merely from using more MLP experts in MoE, we compare our DEFT with the original GATES and a stronger GATES with a deeper MLP-PMM. Table 3 shows that GATES and GATES+deep-MLP PMM achieve nearly identical performance, whereas DEFT clearly outperforms both across all scenarios. This indicates that the improvement stems from MoE's ability to select specialized policies per decision step, rather than from simply increasing network capacity.

**Inference Overhead.** Table 3 also reports the per-step inference overhead on all testing scales. As expected, the original GATES model is the fastest overall, because it does not include any MoE component and therefore avoids extra routing computation. Adding an MoE on top of GATES (all DEFT variants) inevitably introduces some overhead, but the increase is small. DEFT with linear gating and DEFT with our graph-adaptive gating are only slightly slower than GATES, while achieving much lower total cost. Among the DEFT variants, linear gating is the cheapest to run; our graph-adaptive gating adds only a small overhead because it only computes attention weights for expert selection (see Appendix B.2); MLP gating is the most expensive because it must learn new embeddings through multiple fully connected layers at every decision step. Finally, GATES + deep MLP-PMM is the slowest method overall, because it infers through a deeper MLP at each decision step.

Overall, these ablations show that adding an MoE-PMM to GATES reliably improves scheduling performance. Our proposed DEFT offers the best performance and moderate latency, benefiting from its lightweight cross-attention design. Additional ablation studies, including sensitivity to the number of experts, Top-$k$ routing, and pre-training $\gamma$, are provided in Appendix H.

## 6 CONCLUSION

This paper presented **DEFT**, the first Mixture-of-Experts architecture for dynamic cloud workflow scheduling. DEFT trains multiple experts specialized for different levels of deadline tightness and employs a **graph-adaptive gating** network that utilizes workflow DAGs, VM states, and more importantly deadline urgency to activate the most appropriate experts. This combination enables DEFT to adapt its scheduling strategy across diverse scenarios with both flexibility and precision. Extensive experiments demonstrated that DEFT can consistently achieve the lowest overall cost, outperforming heuristic and state-of-the-art DRL baselines by delivering a superior balance between VM rental fees and deadline penalties.

Looking ahead, future research may extend DEFT to multi-tenant cloud environments, integrate explainability into expert routing, and explore broader applications of graph-adaptive MoE models for large-scale resource management.

## ETHICS STATEMENT

Our research focuses on dynamic workflow scheduling in cloud computing environments. The work does not involve human subjects, personally identifiable information, or sensitive data. The experiments are conducted entirely on synthetic and publicly available benchmark datasets. Potential societal impacts are positive, as the proposed methods improve energy and cost efficiency in cloud systems. We do not foresee any negative ethical implications from this research.

## REPRODUCIBILITY STATEMENT

To ensure reproducibility, we provide: (1) a full description of the problem formulation and algorithm design in Sections 3 and 4; (2) detailed experimental settings, including datasets, baselines, and hyperparameters, in Section 5 and Appendix F; and (3) complete pseudo-code for the training algorithm in Appendices C and D. The source code and scripts for reproducing all experiments are publicly available at https://github.com/yashenCS/DEFT.

## LLM USAGE STATEMENT

During manuscript preparation, we used a large language model to assist with language polishing, grammar correction, and rephrasing. We carefully reviewed and edited the LLM-generated text to ensure accuracy and originality. LLM was never used to generate experimental results, algorithm designs, neural network architectures, or other core technical contributions.

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

## A  DETAILED PROBLEM FORMULATION

This section provides a more detailed description of the Cost-Aware Dynamic Workflow Scheduling (CADWS) problem, complementing the concise formulation presented in Section 3.

### A.1  WORKFLOW MODEL

A workflow $W_i \in \mathcal{W}$ is represented by a directed acyclic graph (DAG) $W_i = (\mathcal{O}_{W_i}, \mathcal{C}_{W_i})$, where $\mathcal{O}_{W_i}$ is the set of tasks and $\mathcal{C}_{W_i}$ is the set of precedence edges. Each directed edge $(O_{ni}, O_{nj}) \in \mathcal{C}_{W_i}$ indicates that task $O_{ni}$ must finish before task $O_{nj}$ can begin. A task becomes a *ready task*, denoted by $O_{n^*}$, once all its predecessors have completed. Each task $O_{ni} \in \mathcal{O}_{W_i}$ has a computational demand $cd_{O_{ni}} \in \mathbb{R}^+$, which measures the amount of work required. Workflows arrive dynamically with an arrival time $a_i$ and a deadline

$$d_i = a_i + \gamma \cdot \text{minMakespan}(W_i), \tag{7}$$

where $\gamma \geq 1$ is the deadline relaxation coefficient, and $\text{minMakespan}(W_i)$ is the minimum execution time achievable if all tasks are processed on the fastest VM without waiting.

The proposed DEFT operates in a concurrent multi-workflow setting where all workflows share the same VM pool. At each decision step, it observes a global state that includes all ready tasks across workflows, VM statuses, and workflow deadlines, so inter-workflow resource contention is treated as part of the scheduling problem itself. The state embedding module and graph-adaptive gating network are both defined on this global state, enabling DEFT to learn how workflows interact and to coordinate their execution without any extra cross-workflow coordination module.

### A.2  CLOUD ENVIRONMENT

The cloud provides a pool of VMs $\mathcal{M} = \{m_1, m_2, \ldots, m_{|\mathcal{M}|}\}$, each characterized by a processing speed $v_j$ and an hourly rental cost $c_j$. VMs can be provisioned on demand without fixed capacity limits, but the cost grows with the number of leased instances. If task $O_{ni}$ is assigned to VM $m_j$, its execution time is:

$$T^{\text{exec}}_{O_{ni}, m_j} = \frac{cd_{O_{ni}}}{v_j}, \tag{8}$$

Let $T^{\text{start}}_{O_{ni}}$ be its start time, then the completion time of $O_{ni}$ is:

$$T^{\text{comp}}_{O_{ni}} = T^{\text{start}}_{O_{ni}} + T^{\text{exec}}_{O_{ni}, m_j}, \tag{9}$$

The completion time of a workflow $W_i$ is the finish time of its last task:

$$\text{CT}(W_i) = \max_{O_{nk} \in \mathcal{O}_{W_i}} T^{\text{comp}}_{O_{nk}}, \tag{10}$$

### A.3  VM RENTAL COST AND SLA PENALTY

At each time step $t$, executing the ready task $O_{n^*}$ on VM $m_j$ incurs a cost:

$$C^{\text{vm}}_t = c_j \cdot \left\lceil \frac{cd_{O_{n^*}}}{v_j \cdot 3600} \right\rceil, \tag{11}$$

and the cumulative VM rental cost across the scheduling horizon $T$ is:

$$C^{\text{vm}}_{[0,T]} = \sum_{t=0}^{T-1} C^{\text{vm}}_t, \tag{12}$$

Each workflow $W_i$ is associated with a penalty if it misses its deadline. The penalty is defined as:

$$C^{\text{sla}}_T(W_i) = \beta \cdot \max\{0, \text{CT}(W_i) - d_i\}, \tag{13}$$

where $\beta$ is the penalty coefficient. The total SLA penalty over all workflows is:

$$C^{\text{sla}}_T(\mathcal{W}) = \sum_{W_i \in \mathcal{W}} C^{\text{sla}}_T(W_i). \tag{14}$$

## B  THE DETAILS OF GRAPH-ADAPTIVE GATING NETWORKS

### B.1  DAG-EMBEDDING LEARNING

Each workflow $W_i$ is represented by a directed acyclic graph (DAG), denoted as $W_i = (\mathcal{O}_{W_i}, \mathcal{C}_{W_i})$, where $\mathcal{O}_{W_i}$ is the set of task nodes and $\mathcal{C}_{W_i}$ is the set of precedence edges. For a task node $O_{ni} \in \mathcal{O}_{W_i}$, let $\mathcal{N}(O_{ni})$ denote the set of its neighboring task nodes in the DAG. The input feature of $O_{ni}$ is represented as $\mathbf{h}_{O_{ni}} \in \mathbb{R}^d$.

To capture dependencies among tasks, we employ a Graph Attention Network (GAT). The attention coefficient from task $O_{ni}$ to one of its neighbors $O_{nj}$ is defined as

$$\alpha_{ij} = \frac{\exp\left(\text{LeakyReLU}(\mathbf{a}^\top[\mathbf{W}\mathbf{h}_{O_{ni}} \,\|\, \mathbf{W}\mathbf{h}_{O_{nj}}])\right)}{\sum_{O_{nk} \in \mathcal{N}(O_{ni})} \exp\left(\text{LeakyReLU}(\mathbf{a}^\top[\mathbf{W}\mathbf{h}_{O_{ni}} \,\|\, \mathbf{W}\mathbf{h}_{O_{nk}}])\right)}, \tag{15}$$

where $\mathbf{W}$ is a learnable transformation matrix, $\mathbf{a}$ is a trainable attention vector, and $\|$ denotes concatenation.

The hidden embedding of node $O_{ni}$ is then obtained by aggregating messages from its neighbors with attention weights:

$$\mathbf{h}'_{O_{ni}} = \sigma\left(\sum_{O_{nj} \in \mathcal{N}(O_{ni})} \alpha_{ij}\, \mathbf{W}\mathbf{h}_{O_{nj}}\right), \tag{16}$$

where $\sigma(\cdot)$ is a non-linear activation function, e.g., ReLU.

Once all nodes in $W_i$ are updated, we compute the workflow-level DAG embedding by applying a mean pooling over the task embeddings:

$$\mathbf{h}_{W_i} = \frac{1}{|\mathcal{O}_{W_i}|} \sum_{O_{ni} \in \mathcal{O}_{W_i}} \mathbf{h}'_{O_{ni}}. \tag{17}$$

The resulting $\mathbf{h}_{W_i} \in \mathbb{R}^H$ serves as a compact representation of the workflow DAG, capturing both task-specific features and structural dependencies among tasks. This embedding is later used by DEFT to inform scheduling decisions.

### B.2  CROSS-ATTENTION FOR EXPERT SELECTION

At each decision step, we rank $E$ parallel experts via a cross-attention mechanism that maps contextual features to per-expert weights. Consider a batch of $N$ actions with embeddings $\mathbf{A} \in \mathbb{R}^{N \times D_{\text{act}}}$. Let $\mathbf{g} \in \mathbb{R}^{1 \times D_{\text{dag}}}$ be the learned DAG embedding, $\mathbf{r} \in \mathbb{R}^{1 \times D_{\text{ready}}}$ the ready-task embedding, and $\gamma \in \mathbb{R}^{1 \times 1}$ the SLA deadline coefficient. We broadcast $(\mathbf{g}, \mathbf{r}, \gamma)$ across the batch and form queries:

$$\mathbf{Q} = W_q[\mathbf{A}; \mathbf{g}; \mathbf{r}; \gamma] \in \mathbb{R}^{N \times d}, \tag{18}$$

where $[\cdot; \cdot]$ denotes concatenation and $W_q$ is a learned projection. Each row of $\mathbf{Q}$ is the query for one VM action.

**Q/K/V in cross attention.** We maintain a learnable token table $T \in \mathbb{R}^{E \times d}$, with one $d$-dimensional token per expert. For each action in the batch, the attention inputs are:

$$Q = \mathbf{q}_n \in \mathbb{R}^{1 \times d}, \qquad K = T \in \mathbb{R}^{E \times d}, \qquad V = T \in \mathbb{R}^{E \times d},$$

In standard self-attention, the attention output is:

$$\text{Attention}(Q, K, V) = \text{softmax}\left(\frac{QK^\top}{\sqrt{d}}\right)V. \tag{19}$$

However, in our gating scenario, we only require the attention *weights* to rank experts, and the multiplication by $V$ (which would produce a new representation) is unnecessary. So, let $\mathbf{q}_n$ be the $n$-th row of $\mathbf{Q}$, the scaled dot-product attention gives:

$$\boldsymbol{\alpha}_n = \text{softmax}\left(\frac{\mathbf{q}_n K^\top}{\sqrt{d}}\right) \in \mathbb{R}^E, \tag{20}$$

where $\boldsymbol{\alpha}_n = [\alpha_{n,1}, \ldots, \alpha_{n,E}] \in \mathbb{R}^E$, and $\alpha_{n,e}$ denotes the weight assigned to expert $e$ for action $n$ ($e = 1, \ldots, E$). Stacking over $N$ actions yields $\boldsymbol{\alpha} \in \mathbb{R}^{N \times E}$.

**Sparse Top-$k$ routing.** For each action $n$, we select the index set of the top-$k$ experts

$$\mathcal{E}_k^{(n)} = \mathrm{TopK}(\boldsymbol{\alpha}_n),$$

and re-normalize the selected components to form mixture weights

$$w_{n,e} = \frac{\alpha_{n,e}}{\sum_{j \in \mathcal{E}_k^{(n)}} \alpha_{n,j}} \quad (e \in \mathcal{E}_k^{(n)}), \qquad \sum_{e \in \mathcal{E}_k^{(n)}} w_{n,e} = 1. \tag{21}$$

The routed output for action $n$ is

$$\sum_{e \in \mathcal{E}_k^{(n)}} w_{n,e} \, f_e(\cdot), \tag{22}$$

where $f_e(\cdot)$ denotes the forward network of expert $e$.

## C    TRAINING OF EACH EXPERT UNDER DIFFERENT DEADLINES

Following existing works (Shen et al., 2024; 2025), we use the OpenAI ES (Salimans et al., 2017) to train each expert under a different workflow urgency to learn specific knowledge under this deadline scenario. OpenAI ES is a population-based optimization technique known for its robustness against hyperparameter sensitivity, insensitivity to the design of reward signals, and suitability for parallel implementation, making it particularly effective for policy optimization tasks in dynamic environments (Salimans et al., 2017; Khadka & Tumer, 2018). The main procedure involves the following key steps:

(1) In each training generation, we first sample a population of $N$ individuals centered around the current policy parameters $\hat{\theta}$ from a Gaussian distribution. Specifically, the parameter vector for individual $i$ is generated as:

$$\theta_i = \hat{\theta} + \sigma \epsilon_i, \quad \epsilon_i \sim N(0, I) \tag{23}$$

(2) Next, we evaluate the fitness $F(\theta_i)$ of each individual parameter $\theta_i$, which is defined as the negative of the total scheduling cost (including VM rental fees and SLA violation penalties) obtained by using the policy network parameterized by $\theta_i$:

$$F(\theta_i) = R(\tau) = -\sum_{t=0}^{T-1} C_t^{\mathrm{vm}} - C_T^{\mathrm{sla}}(\mathcal{W}) \tag{24}$$

(3) Subsequently, we update the current policy parameter $\hat{\theta}$ by estimating the gradient to maximize the expected fitness of the population, thereby minimizing the total scheduling cost:

$$\nabla_{\hat{\theta}} E_{\epsilon_i \sim N(0,I)}[F(\hat{\theta} + \sigma \epsilon_i)] \approx \frac{1}{N\sigma} \sum_{i=1}^{N} F(\hat{\theta} + \sigma \epsilon_i) \epsilon_i \tag{25}$$

This process of sampling, evaluating, and updating parameters repeats until a maximum number of generations is reached. The training procedure is outlined in Algorithm 1.

## D    TRAINING OF GRAPH-ADAPTIVE GATING NETWORK

Let $\{\phi_k\}_{k=1}^K$ be the pre-trained MLP experts obtained in Appendix C, where $K$ denotes the number of experts. During the second-stage training, we jointly optimize the graph-adaptive gating network with parameters $\theta^g$, the State Embedding Module (SEM) with parameters $\theta^s$, and all pre-trained experts $\{\phi_k\}_{k=1}^K$. We pack all trainable parameters into a single vector:

$$\hat{\Theta} \leftarrow [\theta^g; \theta^s; \{\phi_k\}_{k=1}^K].$$

---

**Algorithm 1** OpenAI ES for policy training

---

**Input**: Population size: $N$, max number of generation: $Gen$, initial parameters of policy $\pi$ with DEFT: $\hat{\theta}$, initial learning rate: $\alpha$, and the Gaussian standard noise deviation: $\sigma$
**Output**: The trained policy $\pi$
 1: **while** the current number of generation $<= Gen$ **do**
 2:     Randomly sample a CADWS training instance: $Pro$.
 3:     **for** each individual ($i$=1,2,...) **in** $N$ **do**
 4:         Sample a $\epsilon_i \sim \mathcal{N}(0, I)$.
 5:         The parameters of $\pi_i$ represented by individual $i$: $\theta_i = \hat{\theta} + \sigma\epsilon_i$
 6:         Evaluate the fitness value of $F(\theta_i)$ using equation 24 based on $Pro$
 7:     **end for**
 8:     Estimate the policy gradient $\nabla_{\hat{\theta}} \mathbb{E}_{\theta_i \sim \mathcal{N}(\hat{\theta}, \sigma^2 I)} F(\theta_i)$ using equation 25.
 9:     Update parameters of $\pi$: $\hat{\theta} \leftarrow \hat{\theta} + \alpha \frac{1}{N\sigma} \sum_{i=1}^{N} \{F(\hat{\theta} + \sigma\epsilon_i)\epsilon_i\}$.
10: **end while**
11: **return** the trained policy $\pi$

---

Given a state $s_t$ (DAG embedding, ready-task features, VM features, and deadline), the SEM produces a context vector $h_t = f_{\theta^s}(s_t)$, the gating network outputs expert weights $\mathbf{w}_t = g_{\theta^g}(s_t, h_t)$ with $\sum_{k=1}^{K} w_{t,k} = 1$, and the DEFT policy mixes expert output as:

$$\pi_{\text{DEFT}}(a_t|s_t) = \sum_{k=1}^{K} w_{t,k} \, \pi_k(a_t|s_t; \phi_k). \tag{26}$$

The fitness is the negative total scheduling cost in equation 24. The pseudo-code is shown in Algorithm 2.

---

**Algorithm 2** DEFT Training (SEM + Gating + Experts via OpenAI ES)

---

**Inputs:** Pre-trained experts $\{\phi_k\}_{k=1}^{K}$ (to be fine-tuned); initial gating params $\theta^g$; initial SEM params $\theta^s$; CADWS training distribution $\mathcal{D}$; population size $N$; ES hyperparameters (learning rate $\alpha$, noise standard deviation $\sigma$, generations $Gen$).
**Output:** Trained $(\theta^g, \theta^s, \{\phi_k\}_{k=1}^{K})$
 1: Define the trainable parameter vector $\hat{\Theta} \leftarrow [\theta^g; \theta^s; \{\phi_k\}_{k=1}^{K}]$
    **Note:** In OpenAI ES, each individual is sampled as $\Theta_i = \hat{\Theta} + \sigma\epsilon_i$ with $\epsilon_i \sim \mathcal{N}(0, I)$; **Algorithm 1** handles sampling and gradient estimation.
 2: Define FITNESS($\Theta_i$):
 3:     Load $\Theta_i \to (\theta^g, \theta^s, \{\phi_k\}_{k=1}^{K})$ into DEFT
 4:     Sample a CADWS instance $Pro \sim \mathcal{D}$
 5:     Roll out $\pi_{\text{DEFT}}$ on $Pro$: for each step $t$, compute $h_t = f_{\theta^s}(s_t)$, $\mathbf{w}_t = g_{\theta^g}(s_t, h_t)$, and
        $\pi_{\text{DEFT}}(a_t|s_t) = \sum_{k=1}^{K} w_{t,k} \, \pi_k(a_t|s_t; \phi_k)$
 6:     Return FITNESS($\Theta_i$) using equation 24
 7: Optimize $\hat{\Theta}$ by invoking **Algorithm 1** with population size $N$, noise $\sigma$, learning rate $\alpha$, generations $Gen$, and FITNESS as the evaluator
 8: **return** the optimized $(\theta^g, \theta^s, \{\phi_k\}_{k=1}^{K})$

---

# E   VM CONFIGURATION AND WORKFLOW PATTERNS

**VM Configuration.** We adopt six Amazon EC2 VM types ranging from `m5.large` to `m5.12xlarge`, varying in computational power and cost. Table 4 lists their specifications.

**Workflow Patterns.** We simulate four workflow types (CyberShake, Montage, Inspiral, SIPHT) with different DAG structures, as shown in Figure 3. The number of tasks per workflow varies with the scale, as shown in Table 5.

Table 4: The configuration of VM instances.

| VM Type | vCPU/Memory (GB) | Cost ($/hour) |
|---|---|---|
| m5.large | 2/8 | 0.096 |
| m5.xlarge | 4/16 | 0.192 |
| m5.2xlarge | 8/32 | 0.384 |
| m5.4xlarge | 16/64 | 0.768 |
| m5.8xlarge | 32/128 | 1.536 |
| m5.12xlarge | 48/192 | 2.304 |

Table 5: Workflow patterns and sizes.

| Scale | CyberShake | Montage | Inspiral/SIPHT |
|---|---|---|---|
| Small | 30 | 25 | 30 |
| Medium | 50 | 50 | 50/60 |
| Large | 100 | 100 | 100 |

## F  ADDITIONAL TRAINING AND TESTING DETAILS

Training and testing use instances of 10 workflows with a Poisson arrival process ($\lambda = 0.01$). Within any single instance, all workflows share the *same* deadline $\gamma$; across instances, $\gamma$ varies by phase. In Stage-1, each expert is pre-trained on S-scale instances at a *fixed* deadline chosen from $\{1.25, 1.75, 2.25, 5.0\}$. In Stage-2, the state embedding network, gating network, and pre-trained expert networks are jointly optimized on S-scale instances where the per-instance deadline is *sampled* from $\{1.0, 1.25, 1.5, 1.75, 2.0, 2.25, 3.0\}$. Although experts are pre-trained at discrete $\gamma$ values, phase-2 training uses a mixed-deadline distribution that exposes all experts and the gating network to various $\gamma$ values (e.g., 1.5, 2.0). Through this phase, the gating network learns a smooth mapping from varied deadlines and state features to suitable expert choices. For intermediate deadlines such as $\gamma = 1.5$ or $\gamma = 2.0$, the gating network does not simply pick the nearest experts, it also considers the current scheduling pressure, deciding whether SLA risk or VM cost should be prioritized. As a result, it adaptively selects between the experts whose behaviors best match the ongoing context.

For testing, we use 30 instances per scale (S/M/L); each instance draws a single deadline from the same sampled set and applies it to all workflows. No parameters are updated during testing. Additionally, we set Top-$k$ to 1, meaning that the gating network selects one expert at each decision step. Top-1 routing enables the gating network to activate the expert whose behavior best fits the current scheduling state, while still switching among different experts over the full trajectory. The benefit of this Top-1 routing behavior is illustrated in Appendices H.1 and I. Table 6 summarizes the detailed configuration of training and testing.

## G  ADDITIONAL EXPERIMENTS ON CONVERGENCE AND STABILITY

**Convergence analysis.** Figure 4 shows the convergence curves of ES-RL, SPN-CWS, GATES, and DEFT on small- and medium-scale workflows. ES-RL exhibits slow and unstable learning, with

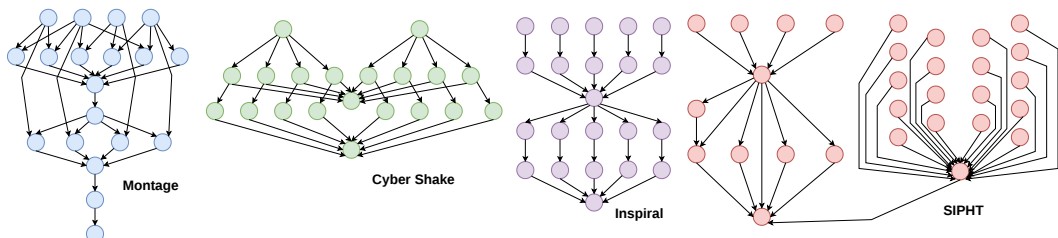

Figure 3: The workflows studied in this work.

Table 6: Training and testing setup. "**Scale**" indicates the workflow scale/size. "**Card.**" stands for cardinality.

| Phase | Updated Params | Scale | $\gamma$ | Card. |
|---|---|---|---|---|
| Stage-1 train | Experts | S | Fixed | $\{1.25, 1.75, 2.25, 5.0\}$ |
| Stage-2 train | SEM+Gate+Exp | S | Sampled | $\{1.0, 1.25, 1.5, 1.75, 2.0, 2.25, 3.0\}$ |
| S test | None | S | Sampled | $\{1.0, 1.25, 1.5, 1.75, 2.0, 2.25, 3.0\}$ |
| M test | None | M | Sampled | $\{1.0, 1.25, 1.5, 1.75, 2.0, 2.25, 3.0\}$ |
| L test | None | L | Sampled | $\{1.0, 1.25, 1.5, 1.75, 2.0, 2.25, 3.0\}$ |

large fluctuations and much lower solution quality in both cases. SPN-CWS converges faster but plateaus at higher total costs, reflecting its limited policy expressiveness. GATES shows more stable convergence and better final performance, yet still lags behind DEFT. In contrast, DEFT not only converges quickly but also reaches the lowest total cost with reduced variance, confirming that its MoE architecture and graph-adaptive gating enable more effective policy learning across different workflow scales.

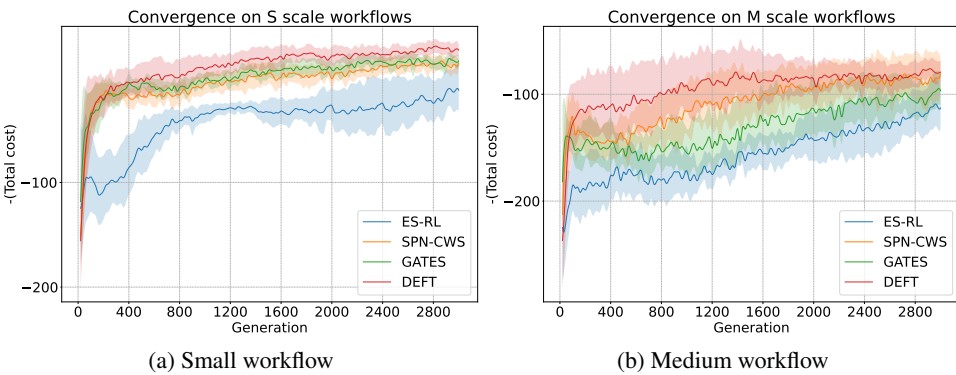

(a) Small workflow        (b) Medium workflow

Figure 4: Convergence of testing performance under dynamic workflow deadlines.

**Performance stability analysis.** Figure 5 compares the distribution of total costs across independent runs on small- and medium-scale workflows. ES-RL exhibits the largest variance, with widely scattered results and frequent extreme outliers, indicating unstable learning behavior. SPN-CWS shows moderate improvement but still suffers from noticeable variability as workflow size increases. GATES achieves tighter distributions with fewer outliers, reflecting more stable scheduling performance. DEFT demonstrates the most concentrated distribution across both scales, with consistently lower variance and narrower interquartile ranges. This highlights that the combination of MoE specialization and graph-adaptive gating not only reduces average total cost but also ensures robust performance stability, which is crucial for real-world deployment.

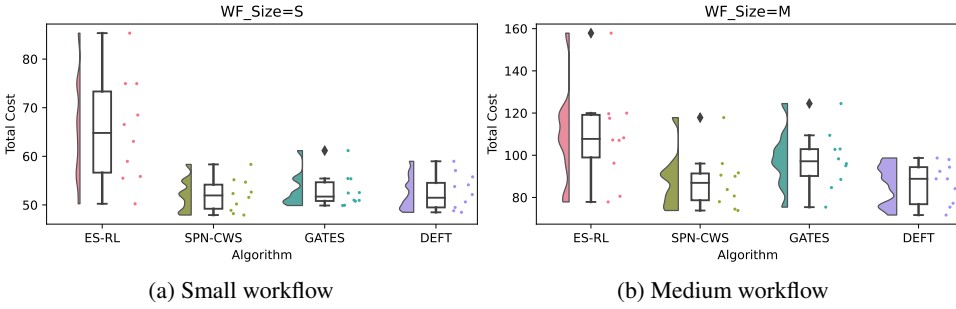

(a) Small workflow        (b) Medium workflow

Figure 5: Stability on the testing set with dynamic workflow deadlines.

# H    Additional Ablation Studies

We present additional ablation studies examining three key parameters in DEFT: (i) the number of experts, (ii) the choice of Top-$k$ routing, and (iii) the selection of $\gamma$ values for expert pretraining. Together, these results verify the architectural choices of DEFT and demonstrate that the proposed graph-adaptive MoE architecture provides consistent and meaningful gains in CADWS. All these ablation experiments are under a small setting with 10 instances, each composed of 5 workflows.

## H.1    Joint Ablation on the Number of Experts and Top-$k$ Routing

To obtain a complete understanding of how the proposed DEFT behaves under different architectural settings in CADWS, we conduct a joint ablation over representative combinations of expert numbers $\{2, 4, 8\}$ and Top-$k$ routing choices $\{1, 2, 4\}$. This results in 8 configurations in total. Table 7 reports the scheduling performance of each configuration.

Table 7: Effect of varying expert counts and Top-$k$ routing choices in DEFT.

| #Experts | Top-$k$ | S | | | M | | | L | | |
|---|---|---|---|---|---|---|---|---|---|---|
| | | Total Cost | VM | SLA | Total Cost | VM | SLA | Total Cost | VM | SLA |
| 2 | 1 | 19.80 | 9.11 | 10.69 | 32.09 | 16.66 | 15.43 | 49.09 | 25.05 | 24.04 |
| | 2 | 20.56 | 9.47 | 11.09 | 35.13 | 18.25 | 16.88 | 53.36 | 26.43 | 26.93 |
| 4 | 1 | 20.18 | 9.17 | 11.01 | **28.32** | 16.73 | 11.59 | **45.20** | 24.04 | 21.16 |
| | 2 | 19.86 | 9.08 | 10.78 | 31.48 | 17.09 | 14.39 | 48.76 | 25.24 | 23.52 |
| | 4 | 20.60 | 9.62 | 10.98 | 30.88 | 16.89 | 13.99 | 48.79 | 25.95 | 22.84 |
| 8 | 1 | **19.61** | 9.20 | 10.41 | 32.67 | 17.08 | 15.59 | 49.02 | 25.25 | 23.77 |
| | 2 | 20.69 | 9.38 | 11.31 | 32.80 | 17.26 | 15.54 | 53.90 | 28.08 | 25.82 |
| | 4 | 20.75 | 9.86 | 10.89 | 33.12 | 17.88 | 15.24 | 50.50 | 28.03 | 22.47 |

**Effect of Expert numbers.** As shown in Table 7, configurations with only two experts consistently underperform on most testing scenarios, particularly in M and L, regardless of the choice of Top-$k$. The limited deadline coverage during pre-training forces the two experts to absorb broad and heterogeneous scheduling patterns, leading to coarse-grained behaviors with insufficient specialization. As a result, the gating network has few meaningful routing options, and increasing $k$ offers no benefit.

Conversely, 8-expert configurations suffer from excessive redundancy. Many pre-training $\gamma$ values are too close, leading experts to converge to nearly identical policies. This redundancy increases routing ambiguity, as multiple experts provide almost the same policy behaviors, making the expert selection harder for the gating network and causing overall performance to degrade across all Top-$k$ settings.

**Effect of Top-$k$ Routing.** Top-1 routing consistently performs best across most settings according to Table 7. Activating only the highest-scoring expert preserves the specialization encoded in each expert policy and avoids the noise introduced by averaging multiple experts' outputs. Increasing $k$ generally weakens this specialization signal and yields diminishing or negative performance, especially when the expert pool already contains overlapping behaviors, as in the 8-expert case.

**4 Experts + Top-1.** The joint ablation reveals a clear and consistent pattern: the number of experts determines how many distinct scheduling policies the DEFT can express, while the Top-$k$ value controls how precisely the gating network of DEFT can leverage that diversity. Their interaction is therefore essential, as experts only perform well under specific Top-$k$ settings and vice versa. Among all eight tested combinations, the configuration with 4 experts and Top-1 routing achieves the strongest overall performance in CADWS. With four experts, the policy pool is diverse enough to include clearly separated SLA-saving and VM-saving strategies, yet compact enough to avoid redundancy. Top-1 routing then allows the gating network to cleanly choose the expert whose policy best matches the current state, resulting in more stable and clearer scheduling decisions. As shown in Table 7, this combination achieves the lowest total cost, especially in M and L testing scenarios. These results indicate that DEFT performs better when expert diversity is meaningful and the gating network can decisively pick the right expert, which is precisely why we adopt the 4-expert Top-1 configuration in this paper.

To further analyze the Top-1 phenomenon, we performed a transparency analysis that logs the VM selection by every expert, as detailed in Appendix I. The results show that Top-1 routing can activate the experts whose policy behaviors best match the current scheduling state, providing better performance than higher Top-$k$ in our CADWS settings.

## H.2   THE CHOICE OF $\gamma$ FOR EXPERT PRE-TRAINING

As the previous ablation in Appendix H.1 has identified the 4-expert Top-1 configuration as the most effective design in DEFT, the following $\gamma$ study is conducted under this setting. We evaluate three configurations of $\gamma$ values for expert pre-training: (1) evenly spaced values, (2) randomly sampled values, and (3) compact cluster values. These $\gamma$-sets differ primarily in how broadly they cover the full spectrum of deadline tightness. Table 8 summarizes the results under the ablation testing scenario.

Table 8: Effect of different $\gamma$ sets for DEFT expert pre-training.

| $\gamma$ set for expert pre-training | S | | | M | | | L | | |
|---|---|---|---|---|---|---|---|---|---|
| | cost | VM | SLA | cost | VM | SLA | cost | VM | SLA |
| Evenly spaced: [1.25, 1.75, 2.25, 5.0] | **20.18** | 9.17 | 11.01 | **28.32** | 16.73 | 11.59 | **45.20** | 24.04 | 21.16 |
| Randomly sampled: [1.25, 1.5, 3.0, 5.0] | 20.61 | 9.67 | 10.94 | 32.70 | 17.92 | 14.78 | 53.41 | 28.70 | 24.71 |
| Compact cluster: [1.0, 1.25, 1.5, 1.75] | 21.04 | 9.21 | 11.83 | 34.39 | 18.09 | 16.30 | 53.49 | 28.16 | 25.33 |

The experiments show a consistent pattern. Both the evenly spaced and the randomly sampled $\gamma$-sets span a wide range of deadline conditions, which exposes each expert to sufficiently different deadline regimes during pre-training. As a result, the experts learn clearly differentiated scheduling styles, from strongly SLA-saving to VM-saving, giving the gating network a diverse and well-separated expert portfolio. This diversity directly translates into better scheduling performance, improving performance compared with the compact cluster set.

In contrast, the manually crafted $\gamma$-set clusters most values in a narrow interval. This restricted coverage causes experts to receive nearly identical training signals, leading them to converge to overly similar policy behaviors. The resulting homogenized expert pool provides little meaningful variation for the gating network to exploit, making expert routing less informative and ultimately degrading scheduling quality.

Overall, these results indicate that DEFT does not require precise tuning of the $\gamma$ values. What matters is simply that the selected $\gamma$-set adequately spans the diversity of deadline tightness levels. Whenever this condition is satisfied, such as with uniformly spaced or randomly sampled values, the expert specialization remains well-structured, and the proposed graph-adaptive gating network can reliably differentiate between experts and select the fittest one at each decision step.

## H.3   SUMMARY

Overall, these ablation studies lead to three clear conclusions about the design of DEFT. First, the MoE architecture of DEFT is effective only when the expert pool provides genuinely diverse scheduling behaviors and the gating network can reliably select among them. Second, the joint ablation over the numbers of experts and Top-$k$ routing shows that this balance is achieved most robustly by the **4-expert Top-**1 configuration in our CADWS problem. Lastly, the choice of pre-training $\gamma$ values does not require careful tuning; DEFT remains stable as long as the selected pre-training $\gamma$-set spans a broad range of all deadline tightness.

## I   THE CHOICE OF TOP-1 ROUTING IN DEFT

From Appendix H.1, we already know that the Top-1 routing in DEFT shows more stable and good performance than higher Top-$k$. To further understand this phenomenon, we performed a transparency experiment that logs the VM selected by each expert. When the current number of VM actions is larger than that in the previous state, the newly appearing actions correspond to newly

rented VMs. This allows us to observe whether an expert prefers renting new VMs or reusing existing ones. We also record the VM occupation rate as an indicator of system load: a high VM occupation rate means the VM pool is heavily occupied with long queues, while a low occupation rate means the system has abundant free VM capacity. Table 9 presents six representative examples from different scheduling states and reveals expert behaviors that explain why Top-1 routing is better suited to CADWS than Top-$k$ ($k > 1$).

Table 9: Transparency experiment. At each decision step, we log the VM chosen by each expert, the current and previous numbers of available VM actions, the current VM occupation rate, and the deadline coefficient $\gamma$.

| Example | VM selection per expert | current VM numbers | previous VM numbers | VM occupation rate | $\gamma$ |
|---|---|---|---|---|---|
| 1 | [28, 28, 28, 28] | 28 | 27 | 0.91 | 1.25 |
| 2 | [47, 47, 47, 38] | 47 | 46 | 0.82 | 2.25 |
| 3 | [25, 8, 8, 48] | 52 | 52 | 0.23 | 1.25 |
| 4 | [6, 26, 18, 21] | 33 | 33 | 0.078 | 2.25 |
| 5 | [34, 24, 34, 25] | 34 | 33 | 0.43 | 1.25 |
| 6 | [54, 62, 48, 33] | 62 | 61 | 0.62 | 2.25 |

## I.1 EXPERT BEHAVIOR ACROSS EXTREME AND TRADE-OFF STATES

Examples 1–4 show two types of extreme scheduling states in which experts tend to converge to a similar decision. Examples 1 and 2 correspond to SLA-critical situations with very high VM occupation rate (0.91 and 0.82). In Example 1, tight deadlines ($\gamma = 1.25$) make reusing existing VMs likely to trigger SLA violations, and all experts select to rent a new VM (index 28). In Example 2, even with a more relaxed deadline ($\gamma = 2.25$), the high VM occupation rate keeps SLA violation risky, and most experts again choose to rent a new VM (47), with only one expert opting for using an existing VM (38). Examples 3 and 4 represent VM-abundant states with low occupation rate (0.23 and 0.078). Under such conditions, the system already has enough idle VM capacity, so renting another new VM tends to increase cost while offering few benefits for avoiding SLA penalties. In these cases, experts naturally agree on reusing existing VMs such as indices 6, 8, 18, or 25, regardless of deadline tightness. These four examples demonstrate that under both extreme states, SLA-critical and VM-abundant states, experts gravitate toward similar or even the same VM action, making Top-1 and Top-$k$ produce nearly identical decisions.

By contrast, Examples 5 and 6 reflect intermediate trade-off states where neither SLA pressure nor VM cost fully dominates. In Example 5, moderate occupation rate (0.43) and a tight deadline ($\gamma = 1.25$) cause experts to split between renting the new VM (34) and reusing existing ones (24 or 25). Example 6 shows similar divergence at a slightly higher occupation rate (0.62) with a relaxed deadline ($\gamma = 2.25$). These trade-off states are exactly where the experts behave differently: some choose the newly rented VM to avoid possible SLA violations, while others reuse existing VMs to keep the VM cost low.

## I.2 WHY TOP-1 IS A BETTER FIT FOR CADWS

In CADWS, the action space is discrete. The scheduler should select one VM at each decision step. This means the gating network cannot "blend" expert recommendations. For example, in Table 10, if Expert 1 strongly prefers VM B and Expert 2 prefers VM C, mixing their action distributions under Top-$k$ can artificially raise the probability of an entirely different VM (e.g., VM D). This blended action distribution can mislead the gating network into picking a VM that none of the experts actually recommends. This limitation has little impact in extreme states (Examples 1–4), where all experts tend to agree on the same VM. However, the situation changes in trade-off states (Examples 5 and 6), where experts genuinely disagree: some prefer renting a new VM to stay safe on SLA violation, while others would rather reuse existing VMs to save VM cost. Top-$k$ ($k > 1$) mixes these conflicting opinions and often blurs the strongest action signal, which can push the scheduler toward a less suitable VM.

The transparency analysis in Table 9 and the examples in Table 10 reveal that: When the system is in an extreme state, Top-1 and Top-$k$ behave almost the same because experts reach natural agreement. When the system enters a trade-off state, Top-$k$ becomes unreliable because it smooths away the

Table 10: Top-$k$ blending can select a VM that no expert actually prefers (expert weights $w_1$=0.4, $w_2$=0.6).

| | Action probability distributions | | | Who prefers this VM? | | |
|---|---|---|---|---|---|---|
| VM | Expert 1 $\pi^{(1)}(a)$ | Expert 2 $\pi^{(2)}(a)$ | Mixed $\pi_{\text{mix}}(a)$ | Expert 1 | Expert 2 | Top-$k$ ($k = 2$) mixing |
| A | 0.10 | 0.05 | 0.07 | – | – | – |
| B | **0.45** | 0.10 | 0.24 | Yes | – | – |
| C | 0.05 | **0.50** | 0.32 | – | Yes | – |
| D | 0.40 | 0.35 | **0.37** | – | – | Yes |

expert differences that actually matter. Top-1 avoids this problem by letting the gating network choose the single expert that best understands the current situation, leading to clearer and more stable decisions, renting VMs when deadlines are tight, and reusing VMs when saving VM cost matters more. This makes Top-1 a better match in our CADWS problem.

### I.3 Expert Selection Across Time

Using Top-1 does not eliminate the benefits of a multi-expert architecture from MoE. In CADWS, a full scheduling process consists of thousands to tens of thousands of decision steps, and the gating network frequently selects among experts in a context-dependent manner. When VM occupation rate rises and workflow deadlines tighten, the gating network chooses towards SLA-saving experts; when deadlines relax, it shifts towards VM-saving experts. Thus, DEFT does leverage diverse expert policies and mixes them across the whole scheduling process, which is exactly how MoE delivers gains in CADWS.

## J Architectural-Level Advances Beyond GATES

Although DEFT reuses the GNN encoder from GATES, it introduces a fundamentally different decision-making mechanism. GATES relies on a single-path priority mapping module (PMM), which restricts the scheduler to one static scheduling mode. In contrast, DEFT replaces this fixed PMM with a MoE architecture by incorporating a graph-adaptive gating network and a set of specialized experts, as shown in Figure 1 (c). The gating network selects the most suitable expert based on the evolving deadline pressure, workflow structure, and VM cost state, enabling DEFT to select policy behaviors dynamically over the whole scheduling horizon. This adaptive, multi-mode expert policy cannot be expressed by GATES. Therefore, DEFT is not a minor modification of GATES but a framework that expands the expressive power of the scheduling policy through expert specialization and deadline-conditioned expert routing.

To further clarify, in our CADWS setting, deadline tightness is the main factor driving scheduling performance, so we use it as the primary dimension for expert specialization in our MoE design. In other scenarios where workflow size, task heterogeneity, or resource type have a stronger impact on scheduling, these attributes could also be used to define expert specializations.

