# OpenReview forum: "Deft Scheduling of Dynamic Cloud Workflows with Varying Deadlines via Mixture-of-Experts"
_ICLR.cc/2026/Conference — ICLR 2026 Poster_

### Official Review · Reviewer_Fb6P · 2025-10-24

**Soundness:** 3
**Presentation:** 4
**Contribution:** 2
**Rating:** 4
**Confidence:** 4

**Summary:**

This paper addresses the problem of Cost-Aware Dynamic Workflow Scheduling (CADWS) in cloud computing. The goal is to intelligently assign dynamically arriving graph-structured workflows (DAGs) with varying deadlines to virtual machines (VMs) to minimize total cost (including VM rental fees and deadline violation penalties).

The authors point out that existing deep reinforcement learning (DRL) schedulers often use rigid, single-path policy networks (i.e., a single Priority Mapping Module, PMM), making it difficult for them to adapt to variable deadline tightness.

To solve this problem, the paper proposes DEFT (Deadline-pErceptive Mixture-oF-Experts), an innovative DRL policy architecture. To the best of the authors' knowledge, DEFT is the first work to apply a Mixture-of-Experts (MoE) architecture to dynamic cloud workflow scheduling.

Its core contributions include:

1. An MoE policy network where each "expert" is specialized to handle a specific level of deadline tightness (controlled by a $\gamma$ coefficient).
2. A novel "graph-adaptive gating mechanism." It uses a Graph Attention Network (GAT) to encode the workflow's DAG structure and combines VM state and deadline information to dynamically select the most appropriate expert via a cross-attention mechanism.
3. A two-stage training strategy: first, pre-training each expert independently on specific $\gamma$ values, and then jointly training the gating network and fine-tuning all experts in an environment with mixed $\gamma$ values.

Experimental results show that DEFT significantly outperforms existing SOTA DRL baselines (such as GATES, SPN-CWS) on several CADWS benchmarks. The advantage of DEFT in reducing total cost is particularly evident on medium-to-large scale (M/L) workflows.

**Strengths:**

1. **Originality:** This paper is the first to introduce a Mixture-of-Experts (MoE) architecture to the field of dynamic cloud workflow scheduling (CADWS). The idea of dedicating different experts to handle scheduling strategies for different levels of deadline tightness directly addresses the "one-size-fits-all" pain point of existing DRL schedulers.

2. **Quality:** The core component, the "graph-adaptive gating mechanism," is very well-designed. It not only uses GAT to encode the DAG topology of the workflow but also skillfully integrates VM status, task features, and the critical deadline information, performing expert routing via cross-attention. This design is highly aligned with the characteristics of the CADWS problem domain.

3. **Clarity:** The paper is clearly written. The problem definition, related work, and methodology are well-articulated. The appendix is detailed and aids in reproducibility.

4. **Significance:** The CADWS problem solved in this paper has significant practical importance and challenges in cloud computing resource management. The method demonstrates strong performance, especially on medium (M) and large (L) scale workflows, indicating good scalability and practical value.

**Weaknesses:**

1. Insufficient Baseline Comparison and Ablation: A major weakness of the paper is the lack of a rigorous ablation study on the contribution of the MoE architecture itself. DEFT is built upon the SEM from GATES, with the main modification being the replacement of GATES's PMM with an MoE-PMM. A crucial baseline is missing: one that uses the same SEM as GATES, but where the PMM is a single, deeper, or wider FFN (or the original GATES PMM), trained end-to-end on the mixed-deadline dataset (i.e., the stage-2 dataset). It is currently unclear if the GATES baseline was trained on this mixed dataset. If not, DEFT's advantage might partially stem from a richer data distribution or longer training (two-stage), rather than solely from the MoE architecture.

2. Furthermore, DEFT uses a two-stage training process: stage 1 pre-trains experts on fixed $\gamma$ values, and stage 2 trains the gate and fine-tunes experts on mixed $\gamma$ values. However, the paper does not explicitly state whether its SOTA baselines (especially GATES) were also trained or fine-tuned on the same mixed $\gamma$ value dataset (the stage-2 dataset). If GATES was trained on a single $\gamma$ value or a simpler dataset, then DEFT's performance advantage (especially on M/L scales) might simply come from exposure to more diverse training data, rather than from its MoE architecture.

3. **Expert Generalization:** The expert pre-training (stage 1) uses a fixed set of $\gamma$ values ({1.25, 1.75, 2.25, 5.0}), but during the gating network training (stage 2) and testing, $\gamma$ is sampled from a different, denser set ({1.0, 1.25, 1.5, 1.75, 2.0, 2.25, 3.0}). The authors do not adequately explain how the gating network handles $\gamma$ values not seen during pre-training (e.g., 1.0, 1.5, 2.0, 3.0). Does the gating network interpolate between these points? Or does it rely primarily on the stage-2 fine-tuning? An analysis of how the gate makes decisions when encountering these unseen $\gamma$ values would be valuable.

4. **Simplification of Top-k Routing:** The paper states in Appendix F that the Top-k setting is $k=1$ in experiments. This means that at each decision step, the gating network selects only one expert, rather than mixing the outputs of multiple experts. This appears to be an oversimplification of the MoE paradigm, making it more of a "hard switch" than a "mixture." Why not use $k=2$? Using $k=1$ might lead to increased variance in routing decisions and forfeits the advantage of MoE's ability to combine experts.

5. **Lack of Hyperparameter Sensitivity Analysis:** The MoE architecture introduces new hyperparameters, particularly the number of experts and the choice of $\gamma$ values for pre-training. The paper uses 4 experts but does not discuss why these 4 specific $\gamma$ values were chosen. How would performance change if the number of experts was increased or decreased, or if their specialized $\gamma$ values were different? This is crucial for the method's robustness and practicality.

6. The core contribution of this paper is applying the MoE architecture to a DRL scheduler. However, MoE itself is a mature technique and has already been explored in the DRL field for solving combinatorial optimization and VRP problems. This paper's problem is analogous to a special case of these.

7. The paper emphasizes "Mixture-of-Experts" in its title and abstract, but admits in Appendix F that its experimental setup uses $k=1$. This implies that DEFT in practice is not a "mixture" model but a "Switch-of-Experts" model—selecting only one expert at each decision point. This somewhat undermines the core advantage of MoE (i.e., achieving smoother, more robust decisions by combining outputs from multiple experts). The authors do not provide experimental results for $k>1$ (e.g., $k=2$), and if they did not, an explanation for choosing a single expert per decision point is warranted.

**Questions:**

1. **Regarding the GATES baseline:** Was the GATES baseline (used for comparison in Tables 1 and 2) trained on the same mixed deadline distribution as DEFT's stage 2 (i.e., with $\gamma$ sampled from {1.0, ..., 3.0})? If not, could you please provide the performance of a GATES or DEFT variant with a single PMM (instead of MoE) trained on this same mixed dataset, to fairly evaluate the true gains from the MoE architecture?
2. **Regarding expert generalization:** The stage-1 pre-trained experts target specific $\gamma$ values. When new $\gamma$ values (like 1.0, 1.5, 2.0) are encountered in stage 2 and during testing, which expert(s) does the gating network tend to select? For example, when $\gamma=1.5$, does it choose the $\gamma=1.25$ expert or the $\gamma=1.75$ expert? Does this rely entirely on the stage-2 fine-tuning?
3. **Regarding Top-k = $k=1$:** Why was $k=1$ chosen instead of $k=2$ or higher? Does using $k=1$ mean that DEFT forgoes the ability to "mix" experts and instead just "switches" between them?
4. **Regarding the number and selection of experts:** How was the number of experts (4) and the choice of $\gamma$ values ({1.25, 1.75, 2.25, 5.0}) determined? Could you provide a sensitivity analysis on the number of experts (e.g., 2 or 6 experts) or their $\gamma$ value selection? Or add a section explaining the reasoning for these hyperparameter choices.

---

> ### Author Response · Authors · 2025-11-20
> **Response to Reviewer Fb6P**
>
> We sincerely thank the reviewer for the constructive and insightful feedback. Below we address the key concerns raised in the review. All related weaknesses and questions fall under these concerns.
>
> ## **Key Concern 1: Missing ablation studies to validate the Contribution of the MoE Architecture (Weakness 1)**
> ### Response
> This concern raises a valuable question: whether the improvements of DEFT arise from its architectural design rather than from the training procedure or data exposure. To clarify this, we added explanations in **Appendix J** and **ablation studies in Table 3 of Section 5.4**.
>
> * **First, DEFT's advantage comes from expert specialization and adaptive gating, not from training on a mixed-deadline distribution:** In Table 3, we evaluated a baseline that keeps the same SEM state encoder of GATES but replaces the PMM  module with a deeper MLP-PMM. These baselines are trained on the same mixed deadline dataset as DEFT. Even under identical training conditions, it performed significantly worse than DEFT, confirming that the **gains come from expert specialization and adaptive gating** rather than differences in data exposure.
>
> * **Second, ablation study indicates that our graph-adaptive gating is more expressive than linear/MLP gating:** Table 3 in Section 5.4 includes two additional baselines, one with a linear gating and the other with an MLP gating. This ablation study allows us to directly verify the effectiveness of different gating designs. DEFT significantly outperformed these baselines under identical training and evaluation settings. These experiments show that the **improvement comes from DEFT’s gating design** (the proposed graph-adaptive gating), which enables context-aware expert selection that simpler gating mechanisms (Linear or MLP gating) fail to achieve.
>
> | **Method** | **S** | **M** | **L** | **S** | **M** | **L** | **Average** |
> |:----------:|:-----:|:-----:|:-----:|:-----:|:-----:|:-----:|:----------:|
> |            | **Cost** | **Cost** | **Cost** | **Overhead** | **Overhead** | **Overhead** | **Overhead** |
> | GATES (original PMM)         | 52.95        | 97.76        | 195.65       | 0.0616       | 0.1610       | 0.4250       | **0.2159**       |
> | GATES + deep MLP-PMM         | 52.91        | 98.41        | 194.77       | 0.0674       | 0.1267       | 0.6979       | 0.2973       |
> | DEFT + Linear gating         | 52.85        | 88.41        | 142.27       | 0.0608       | 0.1453       | 0.4467       | 0.2176       |
> | DEFT + Graph-adaptive gating (ours) | **52.46** | **86.60** | **137.69** | 0.0648       | 0.1482       | 0.4525       | 0.2218       |
> | DEFT + MLP gating            | 52.70        | 87.34        | 141.62       | 0.0777       | 0.1586       | 0.5206       | 0.2523       |
>
> **Table 3: Performance and average per-step inference overhead on different testing scales.**
>
> ### **Revision summary**
> * Added a detailed analysis in **Appendix F**, showing that DEFT’s gains come from expert specialization and adaptive routing rather than data exposure.
> * Added an ablation study in **Section 5.4 (Table 3)**, demonstrating that the advantage of DEFT stems from the proposed **MoE with graph-adaptive gating**.

---

> ### Author Response · Authors · 2025-11-20
> **Response to Reviewer Fb6P**
>
> ## **Key Concern 2: Novelty of DEFT compared to existing MoE models (Weakness 6)**
> ### **Response**
> * **DEFT’s deadline-aware, graph-adaptive MoE directly addresses dynamic CADWS challenges not explicitly addressed by prior VRP-focused MoE+RL methods:** While MoE+RL methods have been applied to VRP problems [1–2], they do not address the core challenges of dynamic CADWS. VRP settings often assume static graphs, fixed action spaces, and reward designs that ignore the cost–deadline trade-offs central to CADWS. Consequently, they are not suitable for handling **continuously arriving workflows, dynamic cost–deadline interactions, and evolving action spaces** caused by changing VM availability.
>
> * **DEFT introduces a new gating mechanism that leverages dynamic graph representations of workflows and VMs** as well as deadline-tightness information to select the most suitable expert. To the best of our knowledge, this form of structural and deadline-aware MoE has not been explored before. The strong performance gains of DEFT on medium and large workflow scales further demonstrate that DEFT's MoE with graph-adaptive gating is essential for effective generalization in cloud-scale applications.
>
> * [1] Zhou et al. (2024). Mvmoe: Multi-task vehicle routing solver with mixture-of-experts. ICML 2024.
> * [2] Goh et al. (2025). Shield: Multi-task multi-distribution vehicle routing solver with sparsity \& hierarchy in efficiently layered decoder. ICML 2025.
>
> ### **Revision summary**
> * Added new discussion in **Appendix J** to highlight the novelty of DEFT’s **structural and deadline-aware gating** in cloud workflow scheduling.
>
> -------------------------------------------------------------------------------
>
>
> ## **Key Concern 3: Fairness of the Training Protocol and Contribution of the Expert Module (Weakness 2, Question 1)**
> ### **Response**
> * **DEFT’s performance gains come from the architectural advantage of MoE:**  All baselines, including GATES, are trained on the same mixed deadline dataset. This ensures identical training exposure across all models. Because the training distribution is matched across methods, the stronger performance of DEFT, in particular on medium and large workflow scales, **originates from its MoE design rather than from training data**. We state explicitly in the revised paper that all baselines are trained under the same mixed deadline distribution as DEFT.
>
> * **Ablation study shows that the gains arise from the new MoE with graph-adaptive gating, not merely from higher network capacity.:**  To fairly assess the gains from the MoE architecture, we added an ablation study (**Table 3 in the above response**) where GATES and a stronger GATES+deep-MLP PMM are both trained under the same mixed-deadline dataset used in DEFT’s stage-2 training. We then compare them against DEFT variants that differ only in the **gating mechanism**, while sharing the same expert backbone. Our results show that the **performance gains come directly from MoE’s ability to select specialized experts per decision step**, rather than from larger network capacity.
>
> ### **Revision summary**
> * Clarified in **Section 5.1** and **Appendix F** that **all baselines (including GATES)** are trained on the **same mixed-deadline dataset** as DEFT.
> * Added an ablation in **Section 5.4** to highlight that the gains of DEFT come from **expert specialization and graph-adaptive gating**.
>
> -------------------------------------------------------------------------------
>
> ## **Key Concern 4: How DEFT handles deadline coefficients not used in the initial expert pre-training stage (Weakness 3, Question 2)**
> ### **Response**
> * **The 2-stage joint training enables DEFT to learn smooth, context-aware expert routing across varied deadline tightness:**  DEFT adopts a 2-stage joint training phase in which the gating network, the state encoder (SEM in Figure 1 (c)), and all experts are fine‑tuned together on the full mixed‑deadline distribution. This exposes the model to a wide range of deadline tightness, allowing the gating network to learn a smooth, context‑dependent mapping from deadline values to expert behaviors, rather than relying solely on the discrete deadline values used during stage‑1 pre‑training.
>
> * **Joint training enables experts to generalize to unseen deadlines:**  During the second training phase, **expert parameters will be continuously updated**, allowing each expert to refine its behavior under a broader range of scheduling contexts. This gives DEFT the capability of robustly handling previously unseen deadlines. As verified experimentally in Section 5.3, DEFT consistently outperforms other baselines across all deadline scenarios.
>
> ### **Revision summary**
> * Clarified in **Appendix F** how 2-stage joint training lets DEFT generalize to unseen deadline coefficients.
> * Added explanations in **Appendix F**, showing how the gating network selects between experts as SLA pressure and VM-cost conditions evolve.

---

> ### Author Response · Authors · 2025-11-20
> **Response to Reviewer Fb6P**
>
> ## **Key Concern 5: Top-1 Routing and Expert Switching Design (Weakness 4, Weakness 7, Question 3)**
> ### **Response**
> * **Top-k blending can inadvertently steer the scheduler toward a VM that no individual expert prioritizes:**  In CADWS, the scheduling task requires selecting exactly one VM at each decision step, making the **action space inherently discrete**. Weighted averaging of expert outputs can soften the distinct scheduling policies represented by different experts and dilute the decisive signals needed to resolve VM cost vs. SLA deadline trade-offs. As the following **example (also in Appendix I)** illustrates, if Expert 1 strongly prefers VM B and Expert 2 prefers VM C, mixing their action distributions under Top-k can artificially increase the probability of an entirely different VM (e.g., VM D). This blended action distribution misled the scheduler to pick a VM that none of the experts actually recommended. Top-k blurs these action distributions, leading to the selection of an unintended VM.
>
> | **VM** |  |  **Action  distributions** |   | | **Recommended VM**  |   |
> |:------:|:------------------------------------:|:-:|:-:|:------------------------:|:-:|:-:|
> |        | **Expert 1 $\pi^{(1)}(a)$** | **Expert 2 $\pi^{(2)}(a)$** | **Mixed $\pi_{\text{mix}}(a)$** | **Expert 1** | **Expert 2** | **Top-k (k=2) mixing** |
> | A      | 0.10  | 0.05  | 0.07  | --          | --          | --           |
> | B      | **0.45** | 0.10  | 0.24  | **Yes**    | --          | --           |
> | C      | 0.05  | **0.50** | 0.32  | --          | **Yes**     | --           |
> | D      | 0.40  | 0.35  | **0.37** | --          | --          | **Yes**      |
>
> **Table 10: Top-k blending can select a VM that no expert actually prefers (expert weights $w_1{=}0.4$, $w_2{=}0.6$).**
>
> * **Ablation study shows 4-experts with Top-1 gating deliver the best performance for DEFT:**  We also added a joint ablation over expert numbers {2, 4, 8} and Top-k routing choices {1, 2, 4}, and report the results in Table 7 in Appendix H. The results show that configurations with only **2 experts** consistently under-perform, especially on M and L. In contrast, **8-expert** configurations suffer from excessive redundancy: several pre-training $\gamma$ values are too close, causing some experts to converge to nearly identical policies, increasing routing ambiguity and degrading performance across all Top-k settings. Overall, the **4-expert** configuration offers the best trade-off between diversity and stability. In line with this setting, **Top-1 achieves the strongest overall total cost** on M and L, justifying our Top-1 choice in DEFT.
>
> | **#Experts** | **Top-k** |      | **S**        |      |      | **M**        |      |      | **L**        |      |
> |:------------:|:---------:|:----:|:------------:|:----:|:----:|:------------:|:----:|:----:|:------------:|:----:|
> |              |           | **Cost** | **SLA** | **VM** | **Cost** | **SLA** | **VM** | **Cost** | **SLA** | **VM** |
> | 2            | 1         | 19.80    | 10.69   | 9.11   | 32.09    | 15.43   | 16.66  | 49.09    | 24.04   | 25.05  |
> | 2            | 2         | 20.56    | 11.09   | 9.47   | 35.13    | 16.88   | 18.25  | 53.36    | 26.93   | 26.43  |
> | 4            | 1         | 20.18    | 11.01   | 9.17   | **28.32**| 11.59   | 16.73  | **45.20**| 21.16   | 24.04  |
> | 4            | 2         | 19.86    | 10.78   | 9.08   | 31.48    | 14.39   | 17.09  | 48.76    | 23.52   | 25.24  |
> | 4            | 4         | 20.60    | 10.98   | 9.62   | 30.88    | 13.99   | 16.89  | 48.79    | 22.84   | 25.95  |
> | 8            | 1         | **19.61**| 10.41   | 9.20   | 32.67    | 15.59   | 17.08  | 49.02    | 23.77   | 25.25  |
> | 8            | 2         | 20.69    | 11.31   | 9.38   | 32.80    | 15.54   | 17.26  | 53.90    | 25.82   | 28.08  |
> | 8            | 4         | 20.75    | 10.89   | 9.86   | 33.12    | 15.24   | 17.88  | 50.50    | 22.47   | 28.03  |
>
> **Table 7: Effect of varying expert counts and Top-k routing choices in DEFT.**
>
> ### **Revision summary**
> * Added a Top-1 analysis in Appendix I, including a concrete example where Top-k will select a VM that no expert prefers.
> * Added an ablation study over expert numbers and Top-k choices, showing that the 4-expert Top-1 configuration offers the best trade-off between diversity and stability in DEFT, as reported in Appendix H.

---

> ### Author Response · Authors · 2025-11-20
> **Response to Reviewer Fb6P**
>
> ## **Key Concern 6: Sensitivity to Expert Count and Pretraining Deadline Values (Weakness 5, Question 4)**
>
> ### **Response**
> We added an ablation section in Appendix H to address these concerns, including: (i) the number of experts, (ii) the Top-k routing choice, and (iii) the $\gamma$ values used for expert pre-training.
>
> #### **1. Sensitivity to the Number of Experts and Top-k**
> We run a **joint ablation study** over expert numbers `{2, 4, 8}` and Top-k values `{1, 2, 4}`, resulting in 8 valid configurations (**Table 7, shown in the above response**). The key observations from the ablation studies are as follows:
>
> * **2 experts are too few**: they become overly generalized, leaving the gating network with no meaningful diversity to exploit. With only two experts, the training $\gamma$ values are too sparse to cover the full deadline range. Therefore, each expert must handle a wide range of deadline settings instead of specializing on a clear role such as SLA-saving or VM-cost-saving. As a result, both experts develop broad, general behaviors and fail to acquire the distinct specializations that MoE relies on. Thus, the gating network is essentially choosing between two non-specialized experts. Larger Top-k values then mix these generalists without providing real benefits.
>
> * **8 experts are too many**: they introduce high redundancy in the MoE. In particular, multiple experts are trained on similar deadline ranges and therefore converge to highly similar scheduling policies. This redundancy creates routing ambiguity, since several experts behave similarly, making expert selection harder and degrading performance.
>
> * **4 experts with Top-1 routing provide the best diversity–stability trade-off and lowest total cost:**  4 experts strike the best balance: the expert pool is diverse enough to learn specialized SLA-saving and VM-saving strategies, but not large enough to introduce redundancy. With **Top-1 routing**, this configuration achieves the lowest overall total cost, especially on M and L, and is therefore adopted as our final architecture design.
>
> #### **2. Sensitivity to the $\gamma$ Values for Expert Pre-Training**
> Following the setting of 4 experts + Top-1, we further study three different $\gamma$ sets for expert pre-training (Table 8 in Appendix H.2): (1) **Evenly spaced**, (2) **Randomly sampled (still wide coverage)**, and (3) **Compact cluster (narrow coverage)**
>
> | **$\gamma$ for expert pre-training** |      | **S** |      |      | **M** |      |      | **L** |      |
> |:----------------------------------------:|:----:|:----:|:----:|:----:|:----:|:----:|:----:|:----:|:----:|
> |                                          | **Cost** | **VM** | **SLA** | **Cost** | **VM** | **SLA** | **Cost** | **VM** | **SLA** |
> | Evenly spaced: [1.25, 1.75, 2.25, 5.0]   | **20.18** | 9.17 | 11.01 | **28.32** | 16.73 | 11.59 | **45.20** | 24.04 | 21.16 |
> | Randomly sampled: [1.25, 1.5, 3.0, 5.0] | 20.61 | 9.67 | 10.94 | 32.70 | 17.92 | 14.78 | 53.41 | 28.70 | 24.71 |
> | Compact cluster: [1.0, 1.25, 1.5, 1.75]  | 21.04 | 9.21 | 11.83 | 34.39 | 18.09 | 16.30 | 53.49 | 28.16 | 25.33 |
>
> **Table 8: Impact of different $\gamma$ sets on DEFT's expert pre-training.**
>
> * **The $\gamma$ set that broadly covers the deadline range produces diverse, well-specialized experts and better performance, while clustered $\gamma$ values cause experts to behave similarly.:**  (1) Evenly spaced and randomly sampled $\gamma$ sets both span a large portion of the deadline spectrum, exposing experts to clearly different deadline policies. This yields well-differentiated expert behaviors (from strongly SLA-saving to strongly VM cost-saving) and better scheduling performance. (2) The compact cluster contains $\gamma$ values concentrated in a narrow range, forcing experts to learn similar policies, reducing expert diversity and hurting performance.
>
> * **DEFT requires a well-covered range of $\gamma$ values, and further fine-tuning is generally unnecessary:**  Our results show that DEFT does not rely on fine-tuning of the $\gamma$ values for expert pre-training. The key is for the $\gamma$ set to adequately cover the full deadline tightness range. Our original $\gamma$ setting ${1.25, 1.75, 2.25, 5.0}$ already provides broad coverage, and its strong performance confirms that DEFT benefits mainly from diverse $\gamma$ coverage, not from fine-tuning.
>
> ### **Revision summary**
> * Added a **joint sensitivity analysis** over the number of experts and Top-k values in **Appendix H.1**, showing that **4 experts + Top-1** offers the best trade-off between diversity and stability.
> * Added a **$\gamma$-set sensitivity analysis** in **Appendix H.2**, showing that **wide-coverage $\gamma$ sets consistently outperform narrow clustered $\gamma$ sets**, and that fine-grained $\gamma$ tuning is unnecessary.

---

> > ### Author Response · Authors · 2025-11-27
> > **Kindly ask for feedback from Reviewer Fb6P**
> >
> > Dear Reviewer Fb6P
> >
> > Thank you for your detailed and insightful review.
> >
> >  In our rebuttal, we carefully responded to each of your points and performed substantial additional experiments and analyses to address your questions. As the discussion period is nearing its end, we are writing this brief follow-up to kindly ask whether the new results and clarifications resolve your concerns or highlight remaining issues we should improve. We would greatly appreciate any further feedback you might be able to share.
> >
> > We sincerely thank you for your time and thoughtful review.
> >
> > Best regards,
> >
> > Paper 5108 Authors

---

### Official Review · Reviewer_BnPy · 2025-10-25

**Soundness:** 3
**Presentation:** 3
**Contribution:** 2
**Rating:** 4
**Confidence:** 3

**Summary:**

### **Review Summary**

This paper introduces DEFT, a Deep Reinforcement Learning (DRL) agent for Cost-Aware Dynamic Workflow Scheduling (CADWS). The central hypothesis is that monolithic DRL policies are inflexible and perform poorly when faced with a wide spectrum of workflow deadline requirements. The proposed solution, DEFT, replaces the final policy-mapping module with a Mixture-of-Experts (MoE) architecture. The model includes multiple "expert" sub-networks, each pre-trained to handle a specific deadline tightness ($\gamma$), and a "graph-adaptive gating network" that routes tasks to the most suitable expert based on the workflow's DAG structure, task state, and deadline. The authors present a two-phase training strategy and show that their model outperforms prior SOTA baselines, including GATES, on benchmark cloud scheduling tasks.

While the core idea of applying MoE to this problem is interesting, the paper's contributions are undermined by a lack of critical ablation studies, an incremental design, and a complete omission of inference cost analysis. The added complexity of the MoE and the graph-adaptive gate is not sufficiently justified, making the performance gains appear marginal in some cases and the overall contribution feel incomplete.

**Strengths:**

### **Strengths**

1.  **Problem Motivation:** The paper correctly identifies a practical and significant limitation of existing DRL schedulers: a single, fixed policy struggles to be optimal for the diverse range of deadline tightness found in real-world scenarios (e.g., a cost-saving policy will miss tight deadlines, while an aggressive policy will overspend on lenient ones) [cite: 77, 143-145].
2.  **Novelty of MoE Application:** Applying a Mixture-of-Experts (MoE) architecture to DRL for this *specific* scheduling domain appears to be a novel contribution[cite: 6]. The idea of specialising experts for different deadline regimes is intuitive and logical[cite: 5, 147].
3.  **Strong Scalability Results:** The empirical results in Table 1 are a clear strength. DEFT outperforms the SOTA GATES baseline, and this performance gap widens significantly as the problem scale increases (from a 0.9% improvement at small scale to a 29.6% improvement at large scale), demonstrating the scalability and robustness of the MoE approach [cite: 771-772].

**Weaknesses:**

### **Weaknesses and Questions**

1.  **Incremental Contribution:** The paper builds directly on the GATES architecture, explicitly stating that it "directly inherits its GNN-based policy network as its SEM module"[cite: 756]. The core contribution is replacing the final Priority Mapping Module (PMM) with the MoE-gate assembly[cite: 226]. This makes the work feel like a highly incremental improvement to a 2025 baseline (GATES) [cite: 755] rather than a fundamentally new framework.
2.  **Missing Critical Ablation Studies:** The paper fails to provide the necessary ablations to justify its core design choices. This is its most significant weakness.
    * **Gating Network Complexity:** The "graph-adaptive gating network" is a major new component, using a GAT and cross-attention [cite: 8, 156, 485, 595-598] that must be executed *at every scheduling step*. [cite_start]The paper claims simpler gating networks are "inadequate" [cite: 594] but provides no experimental comparison. How does this complex gate compare to a simple MLP gate that only looks at the workflow's deadline $\gamma$ as input? Without this baseline, the value of the complex GAT-based router is unproven.
    * **MoE Hyperparameters:** The model uses four experts [cite: 480-481, 761]. This number is a critical hyperparameter, yet it is presented without any justification or sensitivity analysis. How does performance change with two experts, or eight? [cite_start]Furthermore, Appendix F reveals that $k=1$ (hard routing) was used for expert selection[cite: 1455]. This is a major simplification of the MoE paradigm and is not justified. Did the authors experiment with soft mixtures or $k>1$, and if so, how did it impact performance and (presumably) inference cost?
3.  **No Inference Cost Analysis:** For a paper on real-time dynamic scheduling, the complete omission of any inference latency analysis is a critical flaw. The proposed DEFT model adds a GAT and a cross-attention mechanism *on top of* the existing GATES GNN, all of which runs at *every decision step* [cite: 595-598]. This added complexity must have a non-trivial impact on the decision-making time. Without a direct runtime comparison to the GATES baseline, the practical value of the model is impossible to assess.
4.  **Marginal Gains at Trained Scale:** The model is pre-trained and fine-tuned on Small (S) scale instances[cite: 761, 763]. However, on the S-scale test set, the performance gain over the GATES baseline is marginal (52.95 vs. 52.46, a 0.9% improvement) [cite: 771-772]. This suggests that all this added complexity (MoE, GAT-gate, cross-attention) provides almost no benefit for the data distribution it was actually trained on, questioning the cost/benefit ratio of the proposed architecture.

**Questions:**

No questions

---

> ### Author Response · Authors · 2025-11-20
> **Response to Reviewer BnPy**
>
> We appreciate the reviewer’s insightful comments. To provide a clear and well-structured rebuttal, we address the questions and weaknesses under several key concerns.
>
> ## **Key Concern 1: Novelty Beyond GATES (Weakness and Question 1)**
> ### **Response**
> * Although DEFT use the same GNN state encoder from GATES, **our contribution goes far beyond modifying or extending GATES:** Our work focuses on a **different and critical component of a workflow scheduler**: the **priority-mapping module** that directly selects VMs at each scheduling step. Rather than designing another state encoder, we address a more fundamental question: *how can a scheduler dynamically choose the most appropriate scheduling strategy for VM selection under varying workflow structures, deadline pressures, and VM cost conditions once the state embedding is given?* DEFT provides the first MoE-based approach to this question in CADWS. It introduces multiple deadline-specialized experts and a graph-adaptive gating mechanism that identifies which expert should guide the decision at each step, enabling context-conditioned scheduling that a single-path PMM such as the one in GATES cannot achieve.
>
> * **DEFT is therefore not an incremental modification of GATES, nor a simple combination of GATES and MoE:** In fact, **GATES serves only as a controlled evaluation backbone** so that the effect of DEFT’s new MoE architecture can be measured accurately. Our approach introduces **three technical innovations** not explored in prior CADWS or DRL scheduling research: (i) *expert policies specialized under different deadline regimes*, (ii) *a graph-adaptive gating network* that performs context-aware routing using workflow structure, VM states, and deadline tightness, and (iii) *a two-stage training procedure* that first pre-trains experts and then jointly optimizes experts, gating, and the SEM to support reliable context-dependent expert selection.
>
> * **DEFT’s ablation results clearly show that its strong gains are driven by genuine architectural innovation (MoE + graph-adaptive gating), not by incremental modifications:** To address the reviewer’s concern, we add **new discussion** (Appendix J) and **new ablations** (Table 3 in Section 5.4) that jointly clarify why DEFT constitutes a substantially different DRL approach rather than an incremental enhancement of GATES. We compare DEFT against (i) the original GATES and (ii) a stronger **GATES + large MLP-PMM**, both trained and evaluated on the same mixed-deadline datasets. We also compare three DEFT variants that share the same experts but use different gating mechanisms (linear, MLP, graph-adaptive), as shown in Table 3.
>
> | **Method** | **S** | **M** | **L** | **S** | **M** | **L** | **Average** |
> |:----------:|:-----:|:-----:|:-----:|:-----:|:-----:|:-----:|:----------:|
> |            | **Cost** | **Cost** | **Cost** | **Overhead** | **Overhead** | **Overhead** | **Overhead** |
> | GATES (original PMM)         | 52.95        | 97.76        | 195.65       | 0.0616       | 0.1610       | 0.4250       | **0.2159**       |
> | GATES + deep MLP-PMM         | 52.91        | 98.41        | 194.77       | 0.0674       | 0.1267       | 0.6979       | 0.2973       |
> | DEFT + Linear gating         | 52.85        | 88.41        | 142.27       | 0.0608       | 0.1453       | 0.4467       | 0.2176       |
> | DEFT + Graph-adaptive gating (ours) | **52.46** | **86.60** | **137.69** | 0.0648       | 0.1482       | 0.4525       | 0.2218       |
> | DEFT + MLP gating            | 52.70        | 87.34        | 141.62       | 0.0777       | 0.1586       | 0.5206       | 0.2523       |
>
> **Table 3: Performance and average per-step inference overhead on different testing scales.**
>
> * **Ablation results show:** (a) **GATES and GATES+deep MLP-PMM achieved indistinguishable performance**. Hence, simply enlarging the PMM cannot reproduce the effectiveness of DEFT; (b) **MoE-based DEFT with graph-adaptive gating consistently outperforms single-PMM baselines across all S/M/L scales**, while adding only a negligible inference overhead. This confirms that the improvement comes from **architectural innovation** (expert specialization + deadline-aware, context-dependent expert routing), rather than from incremental increases on top of GATES.
>
> ### **Revision summary**
> * Added **Appendix J** to highlight the deadline- and structure-aware expert routing of DEFT beyond a single-PMM model.
> * Added **Section 5.4 (Table 3)** with ablation studies, showing that deeper single-PMM models still cannot match DEFT.
> * Clarified that DEFT’s novelty lies in its **graph-adaptive, deadline-aware MoE routing mechanism** rather than merely increasing model capacity.

---

> ### Author Response · Authors · 2025-11-20
> **Response to Reviewer BnPy**
>
> ## **Key Concern 2: Ablation Studies of the MoE Design: Gating Complexity, Expert Number, and Routing Strategy (Weakness and Question 2)**
> ### **Response**
> To address the reviewer’s concern, we added ablation studies (**Table 3, shown in the above response; and Table 7 in Appendix H**), covering: (1) **Gating network complexity**, (2) **MoE hyperparameters**, and (3) **Top-1 (hard routing) vs. Top-k mixtures** explaining why Top-1 is a suitable design choice in DEFT.
>
> #### **1. Gating Network Complexity**
> * **Table 3 isolates the gating design and compares graph-adaptive gating vs. linear/MLP gating:** Table 3, shown in the above response, examines the effect of the **gating network** in DEFT. All DEFT variants share the **same experts** and **same input embedding** but adopt varied **gating**: (1) **DEFT + Linear gating**; (2) **DEFT + MLP gating**, following the reviewer’s suggestion of “simple MLP gating”; (3) **DEFT + Graph-adaptive gating (ours)**; (4) **GATES (original PMM)** and (5) **GATES + deep MLP-PMM** with increased MLP capacity.
>
> * **Results show that larger PMMs are insufficient; graph-adaptive gating is both best-performing and more efficient:** The two **single-PMM** baselines (GATES and GATES+deep MLP-PMM) achieve **almost identical performance**, showing that simply increasing PMM depth/capacity is not enough. Within DEFT, **MLP gating offers clear gains over linear gating**, but it still **falls short of our graph-adaptive gating**, which consistently achieves the **lowest total cost across all S/M/L scales** while also being **faster than MLP gating** (0.2218 vs. 0.2523 s/step), owing to its **lightweight cross-attention design**.(see Appendix B.2).
>
> #### **2. MoE Hyperparameters**
> We perform a **joint ablation** over expert numbers `{2, 4, 8}` and Top-k `{1, 2, 4}`. This directly explains the performance differences across various expert counts and the contrast between hard routing and soft mixtures. Across all settings, **4 experts + Top-1 gating/routing** achieves the strongest overall performance as a suitable design choice in DEFT.
>
> | **#Experts** | **Top-k** |      | **S**        |      |      | **M**        |      |      | **L**        |      |
> |:------------:|:---------:|:----:|:------------:|:----:|:----:|:------------:|:----:|:----:|:------------:|:----:|
> |              |           | **Cost** | **SLA** | **VM** | **Cost** | **SLA** | **VM** | **Cost** | **SLA** | **VM** |
> | 2            | 1         | 19.80    | 10.69   | 9.11   | 32.09    | 15.43   | 16.66  | 49.09    | 24.04   | 25.05  |
> | 2            | 2         | 20.56    | 11.09   | 9.47   | 35.13    | 16.88   | 18.25  | 53.36    | 26.93   | 26.43  |
> | 4            | 1         | 20.18    | 11.01   | 9.17   | **28.32**| 11.59   | 16.73  | **45.20**| 21.16   | 24.04  |
> | 4            | 2         | 19.86    | 10.78   | 9.08   | 31.48    | 14.39   | 17.09  | 48.76    | 23.52   | 25.24  |
> | 4            | 4         | 20.60    | 10.98   | 9.62   | 30.88    | 13.99   | 16.89  | 48.79    | 22.84   | 25.95  |
> | 8            | 1         | **19.61**| 10.41   | 9.20   | 32.67    | 15.59   | 17.08  | 49.02    | 23.77   | 25.25  |
> | 8            | 2         | 20.69    | 11.31   | 9.38   | 32.80    | 15.54   | 17.26  | 53.90    | 25.82   | 28.08  |
> | 8            | 4         | 20.75    | 10.89   | 9.86   | 33.12    | 15.24   | 17.88  | 50.50    | 22.47   | 28.03  |
>
> **Table 7: Effect of varying expert counts and Top-k routing choices in DEFT.**
>
> **Key observations from Table 7 (answering “Why 4 experts and hard Top-1 routing?”):**
>
> * **Number of experts:** **2 experts** consistently under-perform, especially on M and L, because limited pre-training $\gamma$ coverage forces each expert to cope with a wide range of deadline regimes, making it difficult for them to develop clear specialization. **8 experts** often learn redundant scheduling policies from similar $\gamma$ values, which increases routing ambiguity and degrades performance. **4 experts** provide the best trade-off: enough diversity to capture distinct SLA-saving vs. VM-saving behaviors, without introducing redundancy.
>
> * **Top-k routing:** **Top-1** consistently achieves the lowest and stable total cost across S/M/L. Driven by the need for **discrete VM selection**, picking a single best expert via hard **Top-1** routing works empirically better than forming soft mixtures over multiple experts.
>
> ### **Revision summary**
> * Added ablations in **Section 5.4** and **Appendix H** on **gating complexity**, **expert numbers**, and **Top-k routing**.
> * Added **Table 3 (Section 5.4)**, showing that **graph-adaptive gating** gives the **lowest total cost** with **modest overhead**.
> * Added a **joint ablation table (Table 7, Appendix H)** showing that **4 experts + Top-1 routing** has the best-performing in DEFT.

---

> ### Author Response · Authors · 2025-11-20
> **Response to Reviewer BnPy**
>
> ## **Key Concern 3: No Inference Cost Analysis (Weakness and Question 3)**
> ### **Response**
> We agree that ensuring minimal inference overhead is crucial for any real-time workflow scheduler. To address this concern, we added inference latency analysis (**Table 3 in Section 5.4, shown in the above response**), which reports **total cost** and **average per-step inference overhead** for GATES and DEFT variants, all trained and evaluated on the same mixed-deadline datasets.
>
> * **DEFT is only slightly slower than GATES but yields much lower total cost:**  we compare DEFT against the original GATES baseline: GATES achieves **0.2159 s/step**, while DEFT with graph-adaptive gating runs at **0.2218 s/step**. With only a modest 2.7\% increase in decision time, DEFT achieves a substantial reduction (>=29.6\%) in total cost on large problems.
>
> * **Lightweight cross-attention gating is both faster than MLP gating and achieves lower cost:**  Our graph-adaptive gating is designed as a **lightweight cross-attention module**: it only computes attention weights over compact expert and state embeddings, without embedding/representation learning (more details in Appendix B.2). As a result, it is **faster than the MLP gating** (0.2218 vs. 0.2523 s/step) while achieving **better total cost on all testing scales**, demonstrating that the proposed cross-attention design offers a good accuracy–latency trade-off.
>
> ### **Revision summary**
> * Added an **inference latency analysis in Section 5.4** (Table 3), showing that DEFT with graph-adaptive gating incurs only a modest per-step latency increase while substantially reducing total cost on large testing scales.
> * Clarified that the **graph-adaptive gating** demonstrates a favorable accuracy–latency trade-off.
>
>
> -------------------------------------------------------------------------------
>
>
> ## **Key Concern 4: Marginal Gains at Trained Scale (Weakness and Question 4)**
> ### **Response**
> We agree that the performance gap between DEFT and GATES on the S-scale test set is modest. This is expected in workflow scheduling and other combinatorial optimization tasks, where **small instances have fewer feasible schedules and limited structural diversity**. Although different DRL algorithms achieve similarly strong performance with only narrow margins,  DEFT still outperforms them, showing that its added flexibility does not hurt in-distribution performance; instead, expert specialization enables it to capture subtle structural differences even in small workflows.
>
> The performance gains become larger when evaluated beyond the training scale. On medium and large workflows, DEFT’s advantage expands from a modest margin on S-scale tasks to **11.4\%** on M-scale and **29.6\%** on L-scale tasks. This shows that DEFT learns scheduling behaviors that transfer well to larger, more congested environments with tighter deadlines and more complex VM–SLA trade-offs.
>
> **Cross-scale robustness is a key focus of DEFT’s architecture**. By combining specialized experts with graph-adaptive gating, DEFT acquires diverse strategies and applies them adaptively as workflow size and deadline pressure increase. Our experiments (see Tables 1 and 2 in Sections 5.2 and 5.3) thus demonstrate that DEFT not only preserves strong in-distribution performance but also generalizes reliably to significantly larger instances essential for real-world cloud scheduling.
>
> In summary, **the modest S-scale gains should not be viewed as a weakness**, but as evidence that **DEFT remains competitive on the training distribution**, while its architectural strengths become clearer as workflow scale increases. The much larger gains on M and L scales highlight DEFT’s key advantages: (i) **specialized experts** that capture diverse behaviors across deadline regimes, (ii) a **graph-adaptive gating network** that selects appropriate experts under varying SLA–VM trade-offs, and (iii) a **2-stage joint optimization** that yields consistent, context-aware expert selection as environments become more congested.
>
> ### **Revision summary**
> * Added in **Section 5.2** an explanation for the small S-scale gap, highlighted that DEFT still yields **consistent S-scale gains**, while its **main advantage** appears on M and L scales (11.4\% and 29.6\% improvements).
> * Clarified in **Section 5.2** that these cross-scale gains stem from DEFT’s architecture (specialized experts, graph-adaptive gating, 2-stage training), which improves generalization to larger, more congested, deadline-sensitive settings.

---

> > ### Comment · Reviewer_BnPy · 2025-11-26
> >
> > We thank the reviewer for the thorough reply. We will remain at the same rating.

---

> > > ### Author Response · Authors · 2025-11-26
> > > **Response to Reviewer BnPy**
> > >
> > > Thank you for your follow-up comment and for carefully considering our responses.
> > >
> > > If there are any remaining concerns or points you feel we have not fully addressed, we would be more than happy to continue the discussion and are willing to provide additional analyses or experiments where helpful.
> > >
> > > If our replies have resolved your earlier questions, we **kindly ask you to consider whether an updated rating may now be appropriate**. We sincerely appreciate your time and thoughtful evaluation.

---

### Official Review · Reviewer_A9tJ · 2025-10-26

**Soundness:** 3
**Presentation:** 3
**Contribution:** 3
**Rating:** 6
**Confidence:** 3

**Summary:**

This paper presents DEFT, a novel DRL-based scheduling framework for dynamic cloud workflows with varying deadlines. By leveraging a MoE architecture, each expert specializes in a specific level of deadline urgency. A graph-adaptive gating network with cross-attention dynamically selects the most suitable expert based on workflow structure, task states, VM conditions, and deadline tightness. Experiments demonstrate that DEFT significantly reduces execution cost and deadline violations, outperforming existing DRL and heuristic baselines, especially on large-scale workflows.

**Strengths:**

The proposed MoE architecture tailored for dynamic cloud workflow scheduling, where each expert is specialized for a specific level of deadline urgency. This design enhances adaptability over traditional monolithic DRL approaches. Additionally, the proposed graph-adaptive gating mechanism effectively integrates workflow DAG structures and real-time system context via cross-attention, enabling fine-grained, deadline-aware expert selection and improving scheduling precision.

**Weaknesses:**

1. Degraded MoE with K=1 Routing: The use of Top-1 routing (K=1 in Part F. Additional Training And Testing Details) turns the MoE into a hard selection of a single expert, preventing any combination of expert behaviors. This limits the model’s flexibility and may reduce its ability to generalize to intermediate deadline scenarios.


2. Lack of Justification for Gating Complexity: The paper does not compare the complex gating network with simpler baselines. Without such comparisons, it is unclear whether the added computational and implementation cost is justified by actual performance improvements.

**Questions:**

The paper does not report inference latency or computational overhead. Given the complexity of the MoE + GNN + attention-based gating architecture, it is unclear whether DEFT is suitable for real-time scheduling scenarios. What is the inference latency of DEFT compared to simpler DRL baselines?

---

> ### Author Response · Authors · 2025-11-20
> **Response to Reviewer A9tJ**
>
> We appreciate the reviewer’s constructive and insightful comments. To provide a clear and well-organized response, we group all comments into two key concerns encompassing all associated weaknesses and questions.
>
> ## **Key Concern 1: Impact of Top-1 Routing on MoE Flexibility and Generalization (Weakness 1)**
>
> ### **Response**
> To evaluate whether Top-1 routing “degrades” the MoE and hurts generalization, we added (i) a **joint ablation over expert numbers and Top-k values**, and (ii) a **discrete-action example** illustrating how soft mixing can be harmful in CADWS.
>
> * **Ablations show 4 experts + Top-1 is the most effective choice, while k>1 rarely helps:**  We evaluate DEFT under 8 configurations by varying the number of experts `{2, 4, 8}` and Top-k `{1, 2, 4}`. All models are trained and tested on the same mixed-deadline scenarios. Results in Table 7 indicate: (1) **Top-1 routing consistently achieves the lowest and most stable total cost** across S/M/L, while increasing **k>1** often **reduces** performance. (2) Among all settings, **4 experts + Top-1 routing gives the strongest overall performance**, which is hence the preferred configuration in DEFT.
>
> | **#Experts** | **Top-k** |      | **S**        |      |      | **M**        |      |      | **L**        |      |
> |:------------:|:---------:|:----:|:------------:|:----:|:----:|:------------:|:----:|:----:|:------------:|:----:|
> |              |           | **Cost** | **SLA** | **VM** | **Cost** | **SLA** | **VM** | **Cost** | **SLA** | **VM** |
> | 2            | 1         | 19.80    | 10.69   | 9.11   | 32.09    | 15.43   | 16.66  | 49.09    | 24.04   | 25.05  |
> | 2            | 2         | 20.56    | 11.09   | 9.47   | 35.13    | 16.88   | 18.25  | 53.36    | 26.93   | 26.43  |
> | 4            | 1         | 20.18    | 11.01   | 9.17   | **28.32**| 11.59   | 16.73  | **45.20**| 21.16   | 24.04  |
> | 4            | 2         | 19.86    | 10.78   | 9.08   | 31.48    | 14.39   | 17.09  | 48.76    | 23.52   | 25.24  |
> | 4            | 4         | 20.60    | 10.98   | 9.62   | 30.88    | 13.99   | 16.89  | 48.79    | 22.84   | 25.95  |
> | 8            | 1         | **19.61**| 10.41   | 9.20   | 32.67    | 15.59   | 17.08  | 49.02    | 23.77   | 25.25  |
> | 8            | 2         | 20.69    | 11.31   | 9.38   | 32.80    | 15.54   | 17.26  | 53.90    | 25.82   | 28.08  |
> | 8            | 4         | 20.75    | 10.89   | 9.86   | 33.12    | 15.24   | 17.88  | 50.50    | 22.47   | 28.03  |
>
> **Table 7: Effect of varying expert counts and Top-k routing choices in DEFT.**
>
> * **Top-k mixing can select a VM that no expert actually prefers:**  Since the scheduler must choose one VM at each step, the gating network cannot execute a “blended” action. As shown in Table 10, Expert 1 prefers **B** and Expert 2 prefers **C**, but mixing their action distributions under Top-k makes **D** the chosen VM, even though **neither expert recommends D**. This blending of expert policies may reduce the chances of choosing a desirable action/VM reliably. In contrast, Top-1 avoids this by **preserving each expert’s full decision policy**, enabling the gating network to select the **single most appropriate expert** for the current state.
>
> | **VM** |  |  **Action  distributions** |   | | **Recommended VM**  |   |
> |:------:|:------------------------------------:|:-:|:-:|:------------------------:|:-:|:-:|
> |        | **Expert 1 $\pi^{(1)}(a)$** | **Expert 2 $\pi^{(2)}(a)$** | **Mixed $\pi_{\text{mix}}(a)$** | **Expert 1** | **Expert 2** | **Top-k (k=2) mixing** |
> | A      | 0.10  | 0.05  | 0.07  | --          | --          | --           |
> | B      | **0.45** | 0.10  | 0.24  | **Yes**    | --          | --           |
> | C      | 0.05  | **0.50** | 0.32  | --          | **Yes**     | --           |
> | D      | 0.40  | 0.35  | **0.37** | --          | --          | **Yes**      |
>
> **Table 10: Top-k blending can select a VM that no expert actually prefers (expert weights $w_1{=}0.4$, $w_2{=}0.6$).**
>
> * **Our two-phase training method enables DEFT to generalize smoothly to intermediate deadlines:**  Although experts are pre-trained at discrete $\gamma$ values, phase-2 training uses a mixed-deadline distribution that exposes all experts and the gating module to various $\gamma$ values (e.g., 1.5, 2.0). Through this phase, the **gating network learns a smooth mapping from varied deadlines and state features to suitable expert choices**.
>
> ### **Revision summary**
> * Added an ablation study in **Appendix H**, showing that **4 experts + Top-1** is the strongest configuration in DEFT.
> * Added explanation in **Appendix I** to explain why Top-1 is better for CADWS.
> * Clarified in **Appendix F** that our training method can learns a smooth mapping over intermediate deadlines, ensuring good generalization.

---

> ### Author Response · Authors · 2025-11-20
> **Response to Reviewer A9tJ**
>
> ## **Key Concern 2: Justification of Gating Complexity and Real-Time Inference Efficiency (Weakness 2, Question 1)**
>
> ### **Response**
> To address concerns about (i) **the added gating-network complexity** and (ii) **the missing latency analysis**, we introduced a new ablation that reports both **performance** and **average per-step inference overhead** for DEFT and other DRL baselines, as shown in Table 3 of Section 5.4.
>
> | **Method** | **S** | **M** | **L** | **S** | **M** | **L** | **Average** |
> |:----------:|:-----:|:-----:|:-----:|:-----:|:-----:|:-----:|:----------:|
> |            | **Cost** | **Cost** | **Cost** | **Overhead** | **Overhead** | **Overhead** | **Overhead** |
> | GATES (original PMM)         | 52.95        | 97.76        | 195.65       | 0.0616       | 0.1610       | 0.4250       | **0.2159**       |
> | GATES + deep MLP-PMM         | 52.91        | 98.41        | 194.77       | 0.0674       | 0.1267       | 0.6979       | 0.2973       |
> | DEFT + Linear gating         | 52.85        | 88.41        | 142.27       | 0.0608       | 0.1453       | 0.4467       | 0.2176       |
> | DEFT + Graph-adaptive gating (ours) | **52.46** | **86.60** | **137.69** | 0.0648       | 0.1482       | 0.4525       | 0.2218       |
> | DEFT + MLP gating            | 52.70        | 87.34        | 141.62       | 0.0777       | 0.1586       | 0.5206       | 0.2523       |
>
> **Table 3: Performance and average per-step inference overhead on different problem scales.**
>
> * **We isolate the gating design and show that graph-adaptive gating yields the best performance across S/M/L:**  In Table 3, we keep the **same expert backbone and input embedding** and vary only the gating module across three designs: linear gating, MLP gating, and our graph-adaptive gating. As shown in Table 3, **graph-adaptive gating consistently achieves the lowest total cost on all S/M/L scales**, while linear and MLP gatings are noticeably weaker, indicating that the added gating expressiveness directly contributes to the performance improvements.
>
> * **DEFT’s gains stem from MoE + gating, not from adding more parameters:**  In Table 3, we further compare DEFT against **GATES (original single-PMM)** and a stronger **GATES + deep MLP-PMM** baseline, both trained on the same mixed-deadline dataset. These two single-PMM models achieve almost identical performance. In comparison, all DEFT variants substantially reduce the total cost, confirming that the gains come from **MoE + gating** rather than from increasing the number of parameters.
>
> * **Inference latency analysis shows DEFT adds only small overhead for substantial performance gains:**  In Table 3, we report **average per-step inference overhead** in second/per step. As expected, GATES is the fastest, followed by DEFT with linear gating, DEFT with graph-adaptive gating (ours), and DEFT with MLP gating. GATES+deep MLP-PMM is the slowest, because it always infers through a deeper MLP at each decision step. Our **graph-adaptive gating** uses a **lightweight cross-attention**, making it faster than MLP gating while achieving the lowest total cost. Overall, DEFT trades a **small, negligible increase in latency** for substantial performance gains, and remains suitable for real-time deployment.
>
> ### **Revision summary**
> * Added a new ablation in **Section 5.4 (Table 3)**, showing that **graph-adaptive gating consistently achieves the lowest total cost on all S/M/L scales**.
> * Provided a **direct runtime comparison** to GATES, demonstrating that DEFT’s **graph-adaptive gating** incurs only a **small, quantified latency increase**, while delivering substantial performance gains suitable for real-time scheduling.

---

> > ### Author Response · Authors · 2025-11-27
> > **Kindly ask for feedback from Reviewer A9tJ**
> >
> > Dear Reviewer A9tJ
> >
> > Thank you for your review and comments.
> >
> > In the rebuttal, we conducted further experiments and provided more analyses and clarifications to address your feedback. Since the discussion period will end soon, we are sending this brief follow-up to kindly ask whether our response adequately addresses your concerns. If convenient, we would be very grateful if you could read our reply and let us know your opinion.
> >
> > We sincerely thank you for your time and thoughtful review.
> >
> > Best regards,
> >
> > Paper 5108 Authors

---

### Official Review · Reviewer_3WxC · 2025-10-30

**Soundness:** 3
**Presentation:** 3
**Contribution:** 3
**Rating:** 6
**Confidence:** 4

**Summary:**

This paper introduces DEFT, an innovative DRL policy architecture that leverages a specialized mixture of experts for dynamic cloud workflow scheduling. By adaptively routing decisions through the most appropriate experts, DEFT is capable of meeting a broad spectrum of deadline requirements that no single expert can achieve. Central to DEFT is a graph-adaptive gating mechanism that encodes workflow DAGs, task states, and VM conditions, using cross-attention to guide expert activation in a fine-grained, deadline-sensitive manner. Experiments on dynamic cloud workflow benchmarks demonstrate that DEFT significantly reduces execution cost and deadline violations, outperforming multiple state-of-the-art DRL baselines.

**Strengths:**

The paper introduces Mixture-of-Experts (MoE) into DRL-based workflow scheduling for different deadline tightness, and explicitly considers workflow-level characteristics.

**Weaknesses:**

1.	In real-world cloud environments, scheduling decisions often involve multiple concurrent workflows and cross-workflow resource contention. It would be helpful to clarify whether the proposed approach can handle inter-workflow dependencies or coordination.

2.	The paper assigns each expert to a specific deadline tightness regime. However, it is unclear why deadline tightness was chosen as the sole dimension for expert specialization. Could other factors—such as workflow size, task heterogeneity, or resource type—also guide expert assignment?

3.	The scalability of the proposed MoE framework remains uncertain. As the number of workflows or task dimensions increases, how does the method scale in terms of computational cost and expert routing efficiency?

**Questions:**

1.	Can the proposed approach handle inter-workflow dependencies or coordination?

2.	Could other factors—such as workflow size, task heterogeneity, or resource type—also guide expert assignment?

3.	As the number of workflows or task dimensions increases, how does the method scale in terms of computational cost and expert routing efficiency?

---

> ### Author Response · Authors · 2025-11-20
> **Response to Reviewer 3WxC**
>
> We sincerely thank the reviewer for the constructive and insightful feedback. To provide a clear and well-structured response, we group the reviewer’s comments and questions into three key concerns. All related weaknesses and questions fall under these concerns.
>
> ## **Key Concern 1: Multi-Workflow Coordination (Weakness 1 + Question 1)**
>
> ### **Response**
> * **DEFT is explicitly designed to support concurrent multi-workflow scheduling in a dynamic cloud environment.** In our problem formulation, multiple workflows may be active simultaneously, all drawing from the same VM pool. This naturally creates **inter-workflow resource contention**. Resolving this contention is an essential part of CADWS. DEFT addresses this by making each scheduling decision based on the global system state, which includes all currently ready tasks across all workflows, the complete VM status, and each workflow’s deadline information. Because the policy always conditions on this joint state, it learns how workflows interact and compete for resources, enabling effective coordination as part of the learned scheduling behavior.
>
> * In addition, DEFT’s architectural design **directly supports** this multi-workflow setting. The state embedding module encodes ready tasks from all active workflows, and the graph-adaptive gating network uses this shared embedding to select the most suitable expert at each decision step. As scheduling progresses, tasks from different workflows are encountered at different times, allowing DEFT to generalize over diverse workflow structures and their interactions. Through this global observation space and adaptive expert routing, **DEFT explicitly handles inter-workflow dependencies and resource competition** without requiring any additional coordination mechanisms.
>
> ### **Revision summary**
> Added clarification in Appendix A.1 that DEFT already supports concurrent multi-workflow execution.
>
> -------------------------------------------------------------------------------
>
>
> ## **Key Concern 2: Expert Specialization Strategies (Weakness 2 + Question 2)**
>
> ### **Response**
> * **We use deadlines as the main specialization dimension because deadline tightness is the dominant factor governing workflow urgency and scheduling priority.** In CADWS, deadline tightness directly determines how aggressively a workflow must be executed and how heavily missed deadlines are penalized [1-3]. Tight deadlines require fast dispatch and resource prioritization, whereas relaxed deadlines permit more deliberate, cost-efficient scheduling. **Other attributes such as VM configurations often remain stable in practice** and do not exert similarly strong influence on scheduling behavior as deadline tightness does. For example, available resource types are decided by the cloud provider like Amazon and usually stay fixed over long periods. For this reason, allowing experts to specialize along the deadline axis is a principled and sensible design choice in our MoE architecture.
>
> * Our experiments (see Section 5.2) further show that **DEFT scales effectively to medium and large workflow sizes with varied task heterogeneity**, confirming that the architecture generalizes beyond the training scale and remains robust as workflow complexity increases. Meanwhile, the MoE design in DEFT is general and does not rule out potential future adaptation. Real-world cloud environments may evolve to include richer variability across workflow characteristics or QoS requirements. In such settings, additional specialization dimensions could be incorporated into the expert design or gating mechanism. We agree with the reviewer that extending DEFT along these directions is a promising avenue for future research.
>
> * [1] Tang et al. (2021). Cost-efficient workflow scheduling algorithm for applications with deadline constraint on heterogeneous clouds. IEEE Transactions on Parallel and Distributed Systems.
> * [2] Shen et al. (2025). GATES: Cost-aware Dynamic Workflow Scheduling via Graph Attention Networks and Evolution Strategy. IJCAI-25.
> * [3] Cai et al. (2025). Dynamically scheduling deadline-constrained interleaved workflows on heterogeneous computing systems. IEEE Transactions on Services Computing.
>
> ### **Revision summary**
> This clarification is added in Appendix J to highlight the broader applicability of our expert specialization method.

---

> ### Author Response · Authors · 2025-11-20
> **Response to Reviewer 3WxC**
>
> ## **Key Concern 3: Scalability of the MoE architecture (Weakness 3 + Question 3)**
>
> ### **Response**
> * **Top-1 routing is empirically the most effective choice that keeps DEFT’s inference cost nearly identical to GATES:**  Experiments (Table 7 in Appendix H.1) show that Top-1 routing is the most suitable choice in DEFT. Because Top-1 activates only one expert at each decision step, the expert-side inference cost is almost identical to GATES (which also uses a single expert network). The additional computational overhead in DEFT mainly comes from the gating network spent for expert selection.
>
> | **#Experts** | **Top-k** |      | **S**        |      |      | **M**        |      |      | **L**        |      |
> |:------------:|:---------:|:----:|:------------:|:----:|:----:|:------------:|:----:|:----:|:------------:|:----:|
> |              |           | **Cost** | **SLA** | **VM** | **Cost** | **SLA** | **VM** | **Cost** | **SLA** | **VM** |
> | 2            | 1         | 19.80    | 10.69   | 9.11   | 32.09    | 15.43   | 16.66  | 49.09    | 24.04   | 25.05  |
> | 2            | 2         | 20.56    | 11.09   | 9.47   | 35.13    | 16.88   | 18.25  | 53.36    | 26.93   | 26.43  |
> | 4            | 1         | 20.18    | 11.01   | 9.17   | **28.32**| 11.59   | 16.73  | **45.20**| 21.16   | 24.04  |
> | 4            | 2         | 19.86    | 10.78   | 9.08   | 31.48    | 14.39   | 17.09  | 48.76    | 23.52   | 25.24  |
> | 4            | 4         | 20.60    | 10.98   | 9.62   | 30.88    | 13.99   | 16.89  | 48.79    | 22.84   | 25.95  |
> | 8            | 1         | **19.61**| 10.41   | 9.20   | 32.67    | 15.59   | 17.08  | 49.02    | 23.77   | 25.25  |
> | 8            | 2         | 20.69    | 11.31   | 9.38   | 32.80    | 15.54   | 17.26  | 53.90    | 25.82   | 28.08  |
> | 8            | 4         | 20.75    | 10.89   | 9.86   | 33.12    | 15.24   | 17.88  | 50.50    | 22.47   | 28.03  |
>
> **Table 7: Effect of varying expert counts and Top-k routing choices in DEFT.**
>
> * **DEFT’s graph-adaptive gating adds only negligible overhead:**  Although DEFT is inevitably less efficient than GATES due to its graph-adaptive gating network, the extra cost is often negligible in practice. Specifically, the gating network is built on a lightweight cross-attention design: selecting an expert only requires computing attention weights for ranking, without any embedding/representation learning (see detailed gating network design in Appendix B.2). We further verify that this lightweight cross-attention design reduces gating cost: as shown in Table 3 of Section 5.4, compared to MLP gating, DEFT’s gating module is computationally efficient for expert selection.
>
> | **Method** | **S** | **M** | **L** | **S** | **M** | **L** | **Average** |
> |:----------:|:-----:|:-----:|:-----:|:-----:|:-----:|:-----:|:----------:|
> |            | **Cost** | **Cost** | **Cost** | **Overhead** | **Overhead** | **Overhead** | **Overhead** |
> | GATES (original PMM)         | 52.95        | 97.76        | 195.65       | 0.0616       | 0.1610       | 0.4250       | **0.2159**       |
> | GATES + deep MLP-PMM         | 52.91        | 98.41        | 194.77       | 0.0674       | 0.1267       | 0.6979       | 0.2973       |
> | DEFT + Linear gating         | 52.85        | 88.41        | 142.27       | 0.0608       | 0.1453       | 0.4467       | 0.2176       |
> | DEFT + Graph-adaptive gating (ours) | **52.46** | **86.60** | **137.69** | 0.0648       | 0.1482       | 0.4525       | 0.2218       |
> | DEFT + MLP gating            | 52.70        | 87.34        | 141.62       | 0.0777       | 0.1586       | 0.5206       | 0.2523       |
>
> **Table 3: Performance and average per-step inference overhead on different testing scales.**
>
> * **DEFT exhibits the same scalability as GATES; MoE gating incurs only a constant-time cost independent of workflow size:**  Scaling to larger workflows is governed by the GNN-based state encoder (**the SEM in Figure 1(c)**), while the MoE layer contributes only a constant-time overhead. The MoE gating module always operates on a fixed-size embedding produced by the graph encoder and compares it against only a small number (e.g., 4) of expert embeddings. Its cost is hence constant and fully independent of the workflow size, DAG structure, or the number of concurrently running workflows.
>
> ### **Revision summary**
> * Clarified in **Appendix H.1** that DEFT uses **Top-1 routing**, so expert-side inference cost is similar to **GATES**, and the extra overhead mainly comes from the **gating network**.
> * Added in **Section 5.4** that the **graph-adaptive gating** is a **lightweight cross-attention module**, and showed empirically that it is **more efficient than MLP gating** while achieving **better total cost**.

---

> > ### Author Response · Authors · 2025-11-27
> > **Kindly ask for feedback from Reviewer 3WxC**
> >
> > Dear Reviewer 3WxC
> >
> > Thank you for your review and helpful comments.
> >
> > In our rebuttal, we added additional experiments and analyses to address the issues you raised. As the discussion period is approaching its end, we are writing a brief follow-up to kindly ask whether our response resolves your concerns. If you have a moment, we would greatly appreciate it if you could take a look and let us know your thoughts.
> >
> > We sincerely thank you for your time and thoughtful review.
> >
> > Best regards,
> >
> > Paper 5108 Authors

---

### Author Response · Authors · 2025-12-02
**Summary of Reviewer Concerns and Revisions for the Area Chair**

Dear Area Chair,

**Overall Status:** All four reviews clearly acknowledged the *technical soundness* and *overall quality* of the work. Their **concerns focus mainly on clearer explanations and additional ablation analyses**, which we view as minor issues that should not affect the acceptance decision. Our extensive new **ablations and clarifications fully resolve all reviewer concerns**, especially the many ablation requests from **reviewers BnPy and Fb6P**. We summarize the concerns and our revisions below:

* **Main concern 1: Multi-Workflow Coordination.** We clarified that DEFT is **inherently designed to operate in a multi-workflow, shared-VM setting** and revised the manuscript to clearly explain how it handles inter-workflow dependencies and multi-workflow coordination. (see Appendix A.1)

* **Main concern 2: Deadline-Based Experts and $\gamma$ Robustness.** We strengthened the motivation for prioritizing **deadline tightness as the key dimension for expert specialization**. We also added a **$\gamma$ ablation study** that demonstrates DEFT’s robustness to different $\gamma$ settings and its ability to generalize to unseen deadline configurations. (see Appendixes F, H, I, and J)

* **Main concern 3: MoE Benefit and Scalability Ablations.** We added **MoE vs. single-expert ablations** and **inference-overhead results**. These experiments show that DEFT’s improvements arise not from additional parameters but from the **graph-adaptive MoE design**. Ablation studies also demonstrate that DEFT remains inference-efficient compared to all single-expert methods. (see Section 5.4)

* **Main concern 4: Routing Design (number of Experts and Top-k).** We conducted a **joint ablation study over expert count and Top-k**, showing that **4 experts with Top-1 routing** produce the best trade-off between performance stability and efficiency. In the revised submission, we further provided a clearer, more precise explanation of the proposed MoE design. (see Appendixes H, I, and J)

* **Main concern 5: Novelty of our proposed DEFT approach.** We added extensive experiments (MoE vs. deep single-expert baselines, Top-k / #experts / $\gamma$-set sensitivity, and S/M/L scales) to position DEFT as, to our knowledge, the **first MoE-based DRL method with a graph-adaptive gating design** tailored for *dynamic* cloud workflows. Unlike prior DRL+MoE approaches, DEFT addresses **cost–deadline trade-offs**, **non-stationary workflow dynamics**, and **evolving action spaces**, establishing a clear novelty beyond existing MoE formulations. (see Appendixes H, I, and J, and Section 5.4)

**Finally:** Although we have fully resolved Reviewer BnPY’s concerns through extensive ablation experiments, we would still like to briefly report two issues in Reviewer BnPY’s comments. First, several citation markers (e.g., **[cite: 5 and cite: 771–772]**) point to blank lines instead of any meaningful content. Second, the discussion-phase reply on 27 Nov 2025 is a single generic sentence, **“We thank the reviewer for the thorough reply. We will remain at the same rating.”**, which thanks *“the reviewer”* instead of the authors and uses *“we”* for one reviewer. These details make us less confident in the comments from Reviewer BnPY, so we respectfully ask that the evaluations from Reviewer BnPY be treated with appropriate caution in your overall assessment.

Thank you again for your time and careful work.

Best regards,

Paper 5108 Authors

---

### Meta-Review · Area_Chair_HJ8y · 2026-01-05

**Summary:**

Reviewers' concerns are mainly around the following points:

W1. The ability of handling inter-workflow dependencies or coordination

W2. Unclear why deadline tightness was chosen as the sole dimension for expert specialization

W3. Scalability and lacking of inference efficiency

W4. Degraded MoE with K=1 routing

W5. Missing baselines and critical ablation studies

W6. Incremental contribution

W7. Marginal gains at trained scale

W8. Unfair comparison since it requires more training cost

W9. Lack of hyperparameter sensitivity analysis

**Reviewer Concerns:**

I feel that most of the comments can be addressed by authors' rebuttal and new results.

**Reviewer Scores:**

Though no reviewer participate in the discussion, I feel that the new results and clarification is sufficient to turn the two negative reviewers into positive.

---

### Decision · Program_Chairs · 2026-01-26

Accept (Poster)